

# Gaps in our understanding of ice-nucleating particle sources exposed by global simulation of the UK climate model

Ross J. Herbert[1], Alberto Sanchez-Marroquin[1, 2], Daniel P. Grosvenor[1, 3], Kirsty J. Pringle[4], Stephen R. Arnold[1], Benjamin J. Murray[1], and Kenneth S. Carslaw[1]

[1]Institute for Climate and Atmospheric Science, School of Earth and Environment, University of Leeds, Leeds, LS2 9JT, UK
[2]Barcelona Supercomputing Center (BSC-CNS), Barcelona, Spain
[3]Met Office Hadley Centre, Exeter, Fitzroy Road, Exeter, Devon, EX1 3PB, UK
[4]Edinburgh Parallel Computing Centre, Bayes Centre, University of Edinburgh, EH8 9BT, UK

**Correspondence:** Ross J. Herbert (R.J.Herbert@leeds.ac.uk)

**Abstract.**

Changes in the availability of a subset of aerosol known as ice-nucleating particles (INPs) can substantially alter cloud microphysical and radiative properties. Despite very large spatial and temporal variability in INP properties, many climate models do not currently represent the link between the global distribution of aerosols and INPs, and primary ice production in clouds. Here we use the UK Earth System Model to simulate the global distribution of dust and marine-sourced INPs over an annual cycle. The model captures the overall spatial and temporal distribution of measured INP concentrations, which is strongly influenced by the world's major mineral dust source regions. A negative bias in simulated versus measured INP concentrations at higher freezing temperatures points to incorrectly defined INP properties or a missing source of INPs. We find that the ability of the model to reproduce measured INP concentrations is greatly improved by representing dust as a mixture of mineralogical and organic ice-nucleating components, as present in many soils. To improve the agreement further, we define an optimized hypothetical parameterization of dust INP activity ($n_s(T)$) as a function of temperature with a logarithmic slope of -0.175 K$^{-1}$, which is much shallower than existing parameterizations (e.g., -0.35 K$^{-1}$ for the K-feldspar data of Harrison et al. (2019)). The results point to a globally important role for an organic component associated with mineral dust.

## 1 Introduction

A subset of atmospheric aerosol particles known as ice-nucleating particles (INPs) have the potential to trigger ice formation when immersed in supercooled water droplets. This process is referred to as immersion-mode heterogeneous ice nucleation and generally operates between 0 °C and about -37 °C (Murray et al., 2012), beyond which homogeneous ice nucleation dominates (Koop and Murray, 2016; Herbert et al., 2015). Such primary ice formation in supercooled clouds is followed by several microphysical processes that can substantially alter the cloud radiative properties as well as the generation of precipitation on regional to global scales (Korolev et al., 2017; Vergara-Temprado et al., 2018b; Hawker et al., 2021; Liu et al., 2012; Vergara-Temprado et al., 2017; Shi and Liu, 2019). The importance of ice nucleation was demonstrated by Vergara-Temprado et al. (2018b) who found that defining a realistic INP population over the Southern Ocean leads to an improved simulation





of the radiative properties of mixed-phase clouds. Estimates of Equilibrium Climate Sensitivity (ECS) have been shown to be dependent on the supercooled liquid cloud fraction (Tan et al., 2016) and are sensitive to the magnitude of the cloud-phase

feedback (Zelinka et al., 2020; Ceppi et al., 2017). Murray et al. (2021) hypothesise a link between the availability of INPs and the magnitude of the cloud-phase feedback, thereby linking the global distribution of INP concentrations to ECS estimates. The importance of ice formation for cloud radiative properties and feedbacks provides an incentive to include INPs as an active component of weather and climate models - i.e., treating ice formation as a process that depends on the ice-nucleating properties of transported aerosol particles.

Laboratory, field, and modelling studies have identified several components of aerosol particles that make them effective INPs (Murray and Liu, 2022; Burrows et al., 2022). Mineral dust is a globally important INP type due to its relatively high ice-nucleating ability and abundance (Augustin-Bauditz et al., 2014; Harrison et al., 2016; Ullrich et al., 2017; Kanji et al., 2017; Vergara-Temprado et al., 2017; Price et al., 2018; Harrison et al., 2019; Chatziparaschos et al., 2023). Most mineral dust is emitted from low-latitude arid source regions such as the Sahara and the Arabian Peninsula (Kok et al., 2021), although

high-latitude cold environments are also potential sources of dust and INPs (Paramonov et al., 2018; Tobo et al., 2019; Sanchez-Marroquin et al., 2020; Meinander et al., 2022; Barr et al., 2023). Most of the mineral dust freezing behaviour below -15 °C can be accounted for by the potassium rich feldspar (K-feldspar) component of dust (Atkinson et al., 2013; Harrison et al., 2016; Vergara-Temprado et al., 2017; Harrison et al., 2019; Price et al., 2018; Harrison et al., 2022). Primary marine organic aerosol (PMOA) internally mixed with sea spray aerosol (SSA) is also a globally important type of INP (Schnell and Vali,

1976; Wilson et al., 2015; McCluskey et al., 2018a; Ickes et al., 2020; Irish et al., 2019; Zhao et al., 2021; Creamean et al., 2018). This marine-sourced INP is present in magnitudes that are comparable to dust INPs in regions far from major dust sources, and is thought to be particularly relevant for the Southern Ocean (Vergara-Temprado et al., 2017, 2018b; McCluskey et al., 2018b, 2019, 2023; Zhao et al., 2021). Other particle types can contribute locally or regionally to the INP population, including primary biological aerosol particles such as fungal fragments, bacteria, or pollen (Pummer et al., 2012; Haga et al.,

2013; Wex et al., 2015). Organic material associated with some mineral dusts, such as agricultural soil or aluvial material from river delta, can also enhance the ice-nucleating ability of these dusts (Tobo et al., 2014; O'Sullivan et al., 2014; Pereira Freitas et al., 2023; Barr et al., 2023; Kawai et al., 2023) and volcanic ash can be transiently important (Steinke et al., 2011; Mangan et al., 2017; Maters et al., 2019, 2020).

Global and regional models have been used to estimate the concentrations and seasonal cycles of INPs from different sources

(Lohmann and Diehl, 2006; Hoose et al., 2010; Burrows et al., 2013; Vergara-Temprado et al., 2017; Hummel et al., 2018; Huang et al., 2018; Vergara-Temprado et al., 2018a; McCluskey et al., 2019; Sanchez-Marroquin et al., 2020; Shi et al., 2022; Kawai et al., 2023; Chatziparaschos et al., 2023; Zhao et al., 2021; McCluskey et al., 2023). The simulated INP distributions have proven useful for understanding the relative importance of different INP types, and in quantifying the impacts of INPs on clouds and climate (Vergara-Temprado et al., 2017; Shi and Liu, 2019; Shi et al., 2022; Chatziparaschos et al., 2023; DeMott

et al., 2010; Hawker et al., 2021; Huang et al., 2018; Hummel et al., 2018; Kawai et al., 2023). Most models simulate dust INP concentrations based on simulated dust fields (Burrows et al., 2013; McCluskey et al., 2019; Shi et al., 2022) combined with laboratory measurements of INP activity of atmospheric dust (Niemand et al., 2012). Improved laboratory understanding





of the specific minerals involved in ice nucleation, such as K-feldspar (Atkinson et al., 2013; Harrison et al., 2019) or quartz (Harrison et al., 2019) has enabled global models of INP concentrations based on these specific minerals (Vergara-Temprado et al., 2017; Chatziparaschos et al., 2023). The marine INP source has been represented by coupling either the simulated PMOA concentration (Vergara-Temprado et al., 2017; Burrows et al., 2013) or SSA surface area (McCluskey et al., 2019, 2018b) to laboratory-based parameterizations that describe their ice-nucleating ability (where environmental samples were analysed in a laboratory).

The Met Office Unified Model (UM) has previously been used at the regional scale to study ice-nucleation. Vergara-Temprado et al. (2018b) focused on cyclonic systems over the Southern Ocean and found that the cloud radiative properties are highly sensitive to the abundance of INPs. However, INPs were prescribed in the regional model based on separate global simulations in an offline aerosol model (Vergara-Temprado et al., 2017). Hawker et al. (2021) also used a regional configuration of the UM and focused on tropical deep convective clouds over the Atlantic Ocean, and found that the definition of the immersion-mode freezing parameterization strongly affected the micro- and macroscale cloud properties. In this study we present a global INP model relevant for immersion-mode freezing of liquid water droplets. Our study builds on the work of Vergara-Temprado et al. (2017) who simulated INPs in the global chemical transport model GLOMAP (Global Model of Aerosol Processes, Mann et al. (2010)). Here we use the global atmosphere configuration GA7.1 (Walters et al., 2019) of the Hadley Centre Global Environment Model (HadGEM). GA7.1 is the atmosphere-only setup of the HadGEM climate model, which includes a version of GLOMAP as part of the UK Chemistry and Aerosol (UKCA) sub-model. Although the aerosol model in this study (GLOMAP) is similar to that used in Vergara-Temprado et al. (2017), the aerosol emissions, transport and removal processes are very different. An expanded global dataset of INP measurements is used to evaluate the model and identify missing INP sources. Our study constitutes a first step towards the full inclusion of the ice-nucleation process in the various Met Office modelling systems for weather and climate.

## 2 Methods

The INP model in this study uses simulated aerosol concentrations generated by the UK Earth System Model version 1 (UKESM1), a climate configuration of the UM comprised of several coupled models detailed below. INP concentrations are calculated from the simulated aerosol fields using surface-area and mass dependent parameterizations of immersion-mode freezing according to the singular description (Vali, 2014). This approach assumes that the time dependence of the ice nucleation process is of second order compared with the temperature dependence, and that ice nucleation occurs instantaneously at certain temperature-dependent active sites distributed on the surface of dust particles or within the volume of SSA . The hypothesis of active sites is supported by experimental evidence (Holden et al., 2019, 2021) and can be used to explain in a reproducible way the freezing behaviour of water induced by most materials (Vali, 2014; DeMott et al., 2018). Therefore, although time-dependence of nucleation is observed in laboratory measurements (Herbert et al., 2014; Knopf et al., 2020), for the vast majority of warm- and mixed-phase clouds the dominant driver for primary ice nucleation is via the cooling associated with vertical ascent.



## 2.1 The UK Earth System Model

UKESM1 is an Earth System Model (Sellar et al., 2020) developed by the UK Met Office and the Natural Environment Research Council (NERC). The physical core of the UKESM1 is the Global Coupled climate model 3.1 (GC3.1) comprised of the Global Atmosphere 7.1 (GA7.1) and Global Land 7.0 (GL7.0) configurations that represent the atmosphere-land-ocean-sea ice interactions (Mulcahy et al., 2018; Walters et al., 2019). We use the atmosphere-only configuration of UKESM1. For these simulations we use a standard Atmospheric Model Intercomparison Project (AMIP) setup, whereby the sea-surface temperatures and sea ice fields are prescribed according to observed 2-D time-series for the year 2010. Other relevant land-surface and ocean-surface variables are prescribed using climatologies for the year 2010, including ocean surface chlorophyll concentrations which are used in our representation of marine-sourced INPs and discussed further in Sect. 2.3. The configuration has an atmospheric resolution of 1.25 ° x 1.875 ° in latitude and longitude (about 135 km) and 85 vertical levels. We perform a free-running simulation (i.e., not nudged to reanalysis meteorology) for 20 years with prescribed fields (e.g., sea-surface temperature, surface properties, non-interactive aerosol emissions) taken from the year 2010 and calculate a mean annual cycle relevant for the present-day climate. The representation of aerosol emissions relevant for this study will be presented in the following sections. Although an evaluation of UKESM1 has been carried out by Mulcahy et al. (2020), given that we are employing the atmosphere-only configuration we present additional model evaluation of aerosol species relevant for this study in Sect. 3.

## 2.2 Representation of dust INPs

We use dust concentrations simulated by the Coupled Large-scale Aerosol Simulator for Studies in Climate (CLASSIC) mineral dust scheme (Woodward, 2011) within the UM. CLASSIC prognostically represents dust in six size sections spanning diameters of 0.0632 to 63.2 μm, with interactive emissions calculated every 30 minutes. Dust emission occurs when the near-surface dust friction velocity exceeds a certain threshold value, which is dependent on the texture of the soil as well as the moisture content of the top layer. In the present study, dust is emitted only from model grid boxes that contain a non-zero fraction of bare soil. We assume that the ice-nucleation ability of the mineral component of soil dusts is determined by the K-feldspar content of the particles. This assumption is valid for the majority of desert dusts found worldwide (Atkinson et al., 2013; Harrison et al., 2022, 2019; Price et al., 2018; Chatziparaschos et al., 2023; Reicher et al., 2018), although it under-estimates the ice-nucleation ability of soil dusts that contain organic ice nucleating material, including agricultural soil dusts (O'Sullivan et al., 2014; Tobo et al., 2014) and high-latitude soil dusts (Tobo et al., 2019; Sanchez-Marroquin et al., 2020; Shi et al., 2022; Meinander et al., 2022; Barr et al., 2023). These organic-mineral containing soil dusts will be discussed in Sect. 5.2. Unlike Vergara-Temprado et al. (2017), we do not simulate the emission of K-feldspar as a distinct tracer. Instead, we simplify the model by assuming that 5 % of the dust mass and surface area is K-feldspar, consistent with mass fractions in atmospheric mineral dust particles (Atkinson et al., 2013; Harrison et al., 2019) and surface samples (Jeong, 2024). Although this mass fraction varies with location (about 1 to 20% Harrison et al. (2019); Nickovic et al. (2012); Harrison et al. (2022); Adebiyi et al. (2023)) and particle size (Tanguy Claquin, 1999; Perlwitz et al., 2015), the error in INP abundance caused by



using a fixed fraction is likely to be within the uncertainty of most INP measurement techniques. The assumed fraction of
K-feldspar is discussed further in Sect. 5.1.

Using modelled dust concentrations, the dust INP number concentration in air ($N_{\mathrm{dustINP}}$) at a specific temperature $T$, in
each bin $i$ is calculated using Eq. 1.

$$N_{\mathrm{dustINP_i}}(T) = N_{\mathrm{dust_i}} f_{\mathrm{KF_i}} (1 - e^{n_s(T)\pi d^2}) \tag{1}$$

In this equation, $N_{\mathrm{dust_i}}$ is the dust number concentration in each size bin (L$^{-1}$ of air), $f_{\mathrm{KF_i}}$ is the fraction of the dust mass
which corresponds to K-feldspar, $n_s(T)$ is the temperature-dependent density of ice-nucleating active sites of pure K-feldspar,
and $d$ is the mean diameter of the bin. We define $n_s(T)$ according to the parameterization of Harrison et al. (2019). The bins
are summed to obtain the total $N_{\mathrm{dustINP}}$ (L$^{-1}$ of air).

## 2.3 Representation of marine-sourced INPs

For aerosol emitted from the ocean surface, two methods have previously been used to calculate INP activity. One is based
on the mass concentration of atmospheric PMOA concentrations (Wilson et al., 2015; Zhao et al., 2021). This approach was
used by Vergara-Temprado et al. (2017), obtaining good agreement with measurements of organic carbon. The reliability of
the approach depends on the PMOA concentrations in the model, which are not well defined by measurements. McCluskey
et al. (2019) used a simpler approach in which the ice-nucleating ability of PMOA is assumed to be proportional to the surface
area of SSA, as defined by the measurements-based parameterization of McCluskey et al. (2018b). A model based on this
assumption agreed well with INP measurements over the Southern Ocean and Mace Head (Ireland) (McCluskey et al., 2019).
In our simulation we use the SSA-based approach to calculate marine-sourced INPs, but test both approaches in Sect. 4.2.3.

For the SSA INPs we use SSA concentrations simulated by the modal version of the Global Model of Aerosol Processes
(GLOMAP-mode). GLOMAP-mode, described in full by Mann et al. (2010) and Mulcahy et al. (2020), is a two-moment
aerosol microphysics scheme that prognostically represents five log-normal aerosol modes ranging from particle diameters of
1 nm to 10 μm. SSA is present in two of the soluble modes (accumulation and coarse) and is internally mixed with sulphate,
black carbon, and organic matter. The $N_{\mathrm{marineINP}}$ concentration is calculated according to Eq. 2.

$$N_{\mathrm{marineINP_i}}(T) = N_{\mathrm{SSA_i}}(1 - e^{n_s(T)\pi d^2}) \tag{2}$$

In order to perform these calculations, we extract the number, mass, standard deviation, and centre of the two aerosol modes
that contain SSA. We then redistribute the SSA into six bins (denoted by $i$) from 0.1 to 10 μm. Using 6 bins reduces the
precision in which the size distribution of the SSA can be described, but this only leads to minor effects on the marine-sourced
INP concentration when compared to the uncertainties in the parameterization of the density of ice-nucleating active sites of
the SSA. Using the number concentration of SSA in each bin, $n_i$, we calculate the INP concentration in each bin using Eq. 2,
where $n_s$ is the density of ice-nucleating active sites given by McCluskey et al. (2018b). The concentration in all the bins is
summed in order to calculate the total $N_{\mathrm{marineINP}}$.



For the PMOA-based approach we use the model emissions of PMOA associated with SSA particles (Mulcahy et al., 2020).
Briefly, the fraction of marine organic content in the SSA is correlated with the presence of chlorophyll in the sea surface,
and negatively correlated with the wind speed at 10 m. The sea-air flux of PMOA is calculated from the flux of SSA and the
fraction of marine organic material in the SSA. Adding a source of PMOA in UKESM1 substantially improved the seasonal
cycle of organic aerosol mass and cloud droplet number concentrations in the Southern Ocean (Mulcahy et al., 2020).

Although PMOA is added to the SSA particles, it is transported as aerosol organic matter along with other organic components
from secondary organic aerosol and primary emissions. In order to separate the PMOA from the rest of the organic
carbon, we therefore ran an additional simulation where the PMOA emissions were switched off. The difference in organic
aerosol mass between the two simulations therefore represents the contribution from PMOA emissions. This method was
successfully applied by Mulcahy et al. (2020) using the UKESM1 to compare the modelled PMOA concentrations to measurements
at Amsterdam Island. However, the organic aerosol components are not always linearly additive because of processes
that influence other sources and sinks of organic aerosol. This results in errors in calculated PMOA primarily in regions where
PMOA makes only a small contribution to total organic aerosol (mainly in the Northern Hemisphere). Because of these issues
we restrict our analysis of PMOA INPs to the Southern Ocean boundary layer, where it is an important INP.

PMOA mass is converted into primary marine organic carbon assuming a mass ratio C/PMOA = 1.0/1.9 (Wilson et al.,
2015). The parameterization given by Wilson et al. (2015) is then used to convert the mass concentration of primary marine
organic carbon into an INP concentration using Eq. 3, where [PMOC] is the mass concentration of primary marine organic
carbon (in grams) and $T$ is temperature (in Kelvin).

$$N_{\mathrm{marineINP,PMOA}}(T) = [\mathrm{PMOC}]\, e^{11.2186-0.44593(T-273.15)} \tag{3}$$

Although the parameterization was developed for temperatures between -6 and -27 °C, we use it in the same range as the
other parameterizations (-10 to -30 °C). This method relates the bulk mass of PMOA with a number of INPs. Given that PMOA
is internally mixed with sea salt in SSA, for each aerosol mode we restrict $N_{\mathrm{marineINP,PMOA}}$ using the PMOA method to the
number concentration of SSA.

## 3  Aerosol model evaluation

In this section we compare the simulated aerosol properties relevant for immersion-mode ice nucleation with in-situ measurements
from ground stations. An in-depth evaluation of aerosol in the UKESM1 was performed by Mulcahy et al. (2020). Using
long-term ground-based observation networks (EMEP, IMPROVE, and EANET) (Mulcahy et al., 2020) demonstrated that
UKESM1 reproduces the global distribution and present-day trends of aerosol from both natural and anthropogenic sources
with reasonable skill. A comparison with multiple satellite products shows good agreement with the spatial distribution and
seasonal cycle of aerosol optical depth, although there is low bias close to dust source regions. Checa-Garcia et al. (2021)
evaluated a number of Earth system models including the UKESM1 with a specific focus on dust. The authors show that





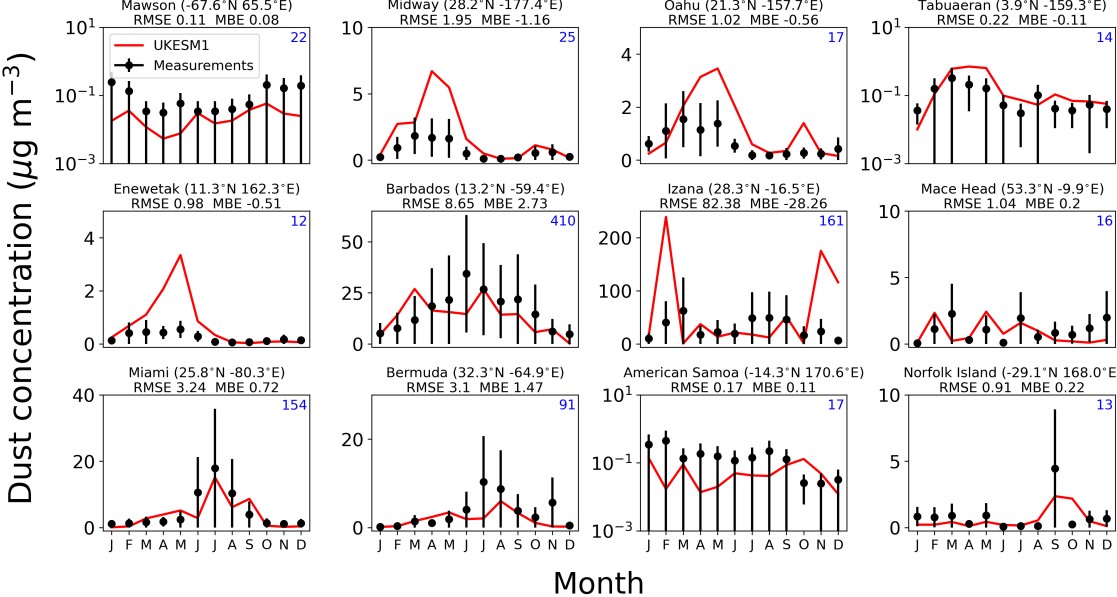

**Figure 1.** Comparison of simulated dust concentrations with measurements. Lines correspond to the simulated monthly mean concentration at the location and dots correspond to the monthly mean measurements from the Huneeus et al. (2011) database at twelve locations with standard deviations. The RMSE and MBE for each location are shown above each plot. The number shown in the top right of each plot is the mean number of measurements per month. Note that all stations have different y-axis scales, with Mawson, Tabuaeran and American Samoa stations shown on a log scale.

the UKESM1 is able to generally capture the observed deposition flux, surface concentrations, and dust loading, but underrepresents the dust optical depth. For our INP model we are primarily interested in representing the dust concentrations so poor dust optical depth performance is of secondary importance to our analysis. We extend these evaluations with some additional focus on the three sources of INP-types relevant for this study: dust (Fig. 1), SSA (Fig. 2), and PMOA (Fig. 3).

The dataset of Huneeus et al. (2011) provides decadal-scale measurements of surface-level dust concentrations from several locations around the globe from the 1980s to the 1990s. Monthly mean measurements are shown alongside monthly mean simulated dust concentrations from the surface level of the model in Fig. 1. Overall, the model reproduces the relative magnitudes of dust concentrations and is able to capture seasonal variability. Model-measurement agreement is also good at remote locations far from dust sources, such as the Antarctic (Mawson) and the South Pacific Ocean (American Samoa and

Norfolk Island). At Norfolk Island there is an interanually variable peak in September which is well reproduced, as are the low magnitudes ($< 10$ μg m$^{-3}$) at the other two stations. The greatest concentrations (20 to 100 μg m$^{-3}$) are measured at Izana, Barbados, Bermuda, and Miami, which are all downwind from African dust source regions (Glaccum and Prospero, 1980; Prospero et al., 1981). Here, the model is able to capture the magnitudes and seasonal variability, though for winter months (November to February the model overpredicts the dust concentrations in Izana by a factor of four. This region, off the




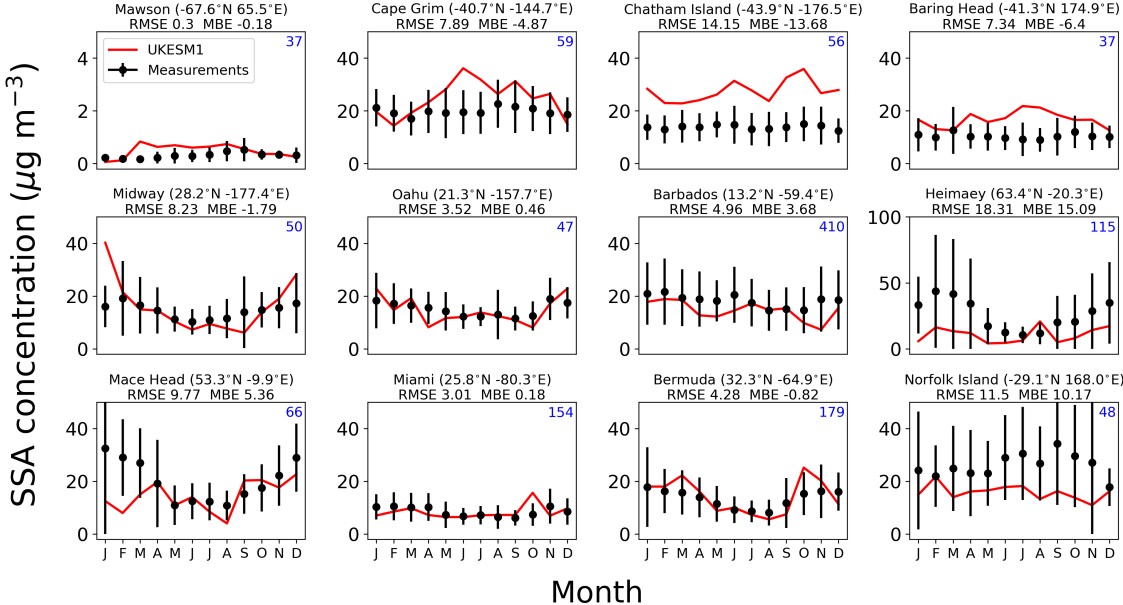

**Figure 2.** Comparison of simulated sea spray aerosol (SSA) concentrations with measurements. Lines correspond to the simulated monthly mean concentration at the location (both accumulation and coarse modes) and dots correspond to monthly mean measurements. The RMSE and MBE for each location are shown above each plot. The number shown in the top right of each plot is the mean number of measurements per month. Note the different y-axis scales used for the Mawson and Heimaey stations.

west coast of Africa, is subject to periodic plumes of dust that are strongly influenced by decadal-scale oscillations during the wintertime (Nakamae and Shiotani, 2013). Additionally our simulation uses some prescribed climatologies for the year 2010 so some discrepancy could be attributed to interannual or multi-decadal variability. A similar overprediction (factor of four) is observed in the tropical North Pacific island of Enewetak, influenced by the long-range transport of dust from Asia (Duce et al., 1980); similar factors could explain the discrepancies. Additional reasons for over prediction are poorly represented dust removal processes and/or the insoluble nature of the simulated dust which would inhibit removal via cloud droplet activation. We revisit the representation and simulation of dust emissions in Sect. 5.

The SSA evaluation is shown in Fig. 2. Monthly mean concentrations of accumulation and coarse modes are compared with historical measurements (1980s to 1990s) from the AEROCE (Atmosphere/Ocean Chemistry Experiment) and SEAREX (Sea/Air Exchange) programs, available from the AEROCOM benchmark data website (https://aerocom.met.no/data, last access: 26 January 2024). Compared to the dust, SSA measurements display considerably less variability amongst the twelve stations, with less pronounced seasonal variation in general. The model is within the range of measurement uncertainty in most stations, although tends to over-predict at Chatham Island and Baring Head (both situated in the region of New Zealand). Although within measurement uncertainty, the model tends to consistently under predict the magnitude at Heimaey (North





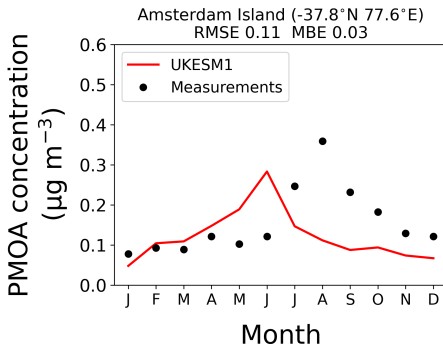

**Figure 3.** Comparison of simulated primary marine organic aerosol (PMOA) concentrations in our simulation with measurements at Amsterdam Island. The line corresponds to the simulated monthly mean concentration at the location and the dots correspond to monthly mean measurements. The RMSE and MBE is shown above the plot.

Atlantic) throughout the annual cycle by up to a factor of four. This is important as this station has the highest concentrations
of all the stations, although also displays the greatest interannual variability, which may account for the model discrepancy.

We compare simulated PMOA concentrations to measurements from Amsterdam Island (southern Indian Ocean) from Sciare et al. (2009) in Fig. 3. The discrepancies between the modelled and measured PMOA are consistent with Mulcahy et al. (2020), who also found that UKESM1 captures the magnitude and seasonality of the PMOA measurements in Amsterdam Island, but with the August peak concentration occurring about two months earlier in the model than in the measurements. This may be
related to an early peak in simulated chlorophyll evident in Mulcahy et al. (2020), which is used to determine the PMOA emission flux. Nonetheless, the comparison suggests the model is able to capture the correct magnitude in PMOA, with the added caveat that there is a distinct lack of long-term PMOA measurements available to evaluate the model.

## 4   The INP model

### 4.1   INP spatial distribution

The annual mean horizontal distribution of simulated INP number concentration from the sum of marine and dust sources ($N_{\mathrm{INP}}=N_{\mathrm{dustINP}}+N_{\mathrm{marineINP}}$) at an altitude of 500 m is shown in Fig. 4. A reference activation temperature of -20 °C is used to calculate $N_{\mathrm{INP}}(T)$, which is the concentration that would be measured in a droplet freezing assay or an online INP counter at -20 °C, or if this air parcel were cooled to -20 °C. Dust INP concentrations (Fig. 4a) are greatest close to the major dust source regions of central and Northern Africa, across much of Asia, and Australia. The outflow of these Northern
Hemisphere (NH) dust sources result in elevated concentrations over the North Atlantic and North Pacific Oceans, whilst the South Pacific and Southern Oceans are influenced by outflow from Australia. There is a clear hemispheric gradient, with about two orders-of-magnitude higher dust INP concentrations in the northern high latitudes than in the south. Marine-sourced INP concentrations (Fig. 4b) are lower and less spatially variable than dust, with $N_{\mathrm{INP}}(-20\ °\mathrm{C})$ around $10^{-3}\ \mathrm{L}^{-1}$ over much of




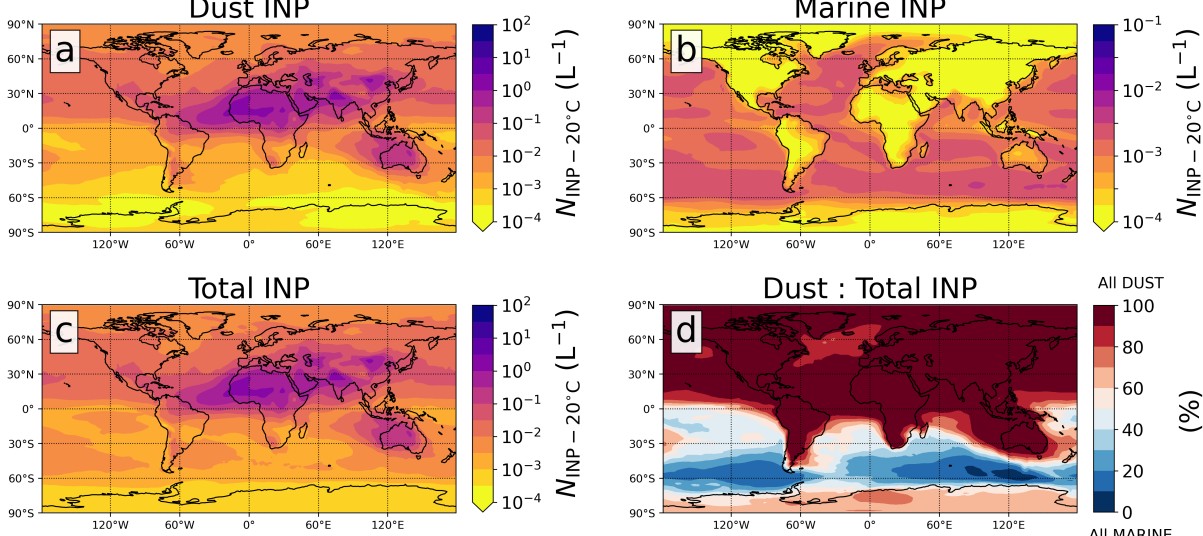

**Figure 4.** Annual mean $N_{\mathrm{INP}}(T)$ concentrations in the boundary layer at 500 m and a temperature of -20 °C. Panel (a) isolates the concentration produced by dust $N_{\mathrm{INP}}(-20\ °\mathrm{C})$, (b) isolates the marine $N_{\mathrm{INP}}(-20\ °\mathrm{C})$ parameterized using SSA, and (c) shows the total $N_{\mathrm{INP}}(-20\ °\mathrm{C})$ from both sources. Panel (d) shows the ratio (expressed as a percentage) of dust $N_{\mathrm{INP}}(-20\ °\mathrm{C})$: total $N_{\mathrm{INP}}(-20\ °\mathrm{C})$, and is used to highlight regions that are dominated by either the dust or marine sources of INPs.

the world's Oceans. The total (dust plus marine-sourced) INP concentrations (Fig. 4c) are dominated by dust in most regions, with marine-sourced INPs making up a substantial fraction over the Southern Ocean (Fig. 4d).

Figure 5 shows the INP number concentration that are active at ambient temperatures, defined as $N_{\mathrm{INP}}(T_{\mathrm{amb}})$. $N_{\mathrm{INP}}(T_{\mathrm{amb}})$ is the INP concentration that we would expect to nucleate ice if activated to cloud droplets. The highest values of $N_{\mathrm{INP}}(T_{\mathrm{amb}})$ occur at high latitudes or high altitudes where temperatures are the lowest. The NH high latitudes have relatively high concentrations throughout the lowest 4 km for most of the year, caused by the long-range transport of dust from the NH low latitude regions. Figures 5a – d show that in the Southern Hemisphere non-negligible concentrations of $N_{\mathrm{INP}}(T_{\mathrm{amb}})$ are only present within the lowest 2 km at latitudes south of 60 °S, with concentrations greater than $10^2\ \mathrm{L}^{-1}$ for most of the year. The very cold lower-tropospheric temperatures present in this region enhance the activity of the very low concentrations of aerosols there that can act as INP. Figures 5e – h show that below 60 °S both marine and dust sources contribute to $N_{\mathrm{INP}}(T_{\mathrm{amb}})$, with marine sources more important over the Southern Ocean, and dust more important over the Antarctic landmass. The largest contribution of marine-sourced INPs occurs during March – May (MAM), while its minimum occurs during December – February (DJF). In contrast, the contribution of marine INP sources to $N_{\mathrm{INP}}(T_{\mathrm{amb}})$ in the NH is negligible for most months of the year, only peaking at less than 5 % between October and March. Here in the NH non-negligible $N_{\mathrm{INP}}(T_{\mathrm{amb}})$ concentrations within the lower troposphere extend as far south as 40 °N for all seasons except June - August (JJA). Although the temperatures in this region are not as cold as in the Southern Hemisphere polar region there is a greater supply of dust from the tropical and subtropical dust sources present in the NH. The development of the Arctic polar dome, evident from SON through to




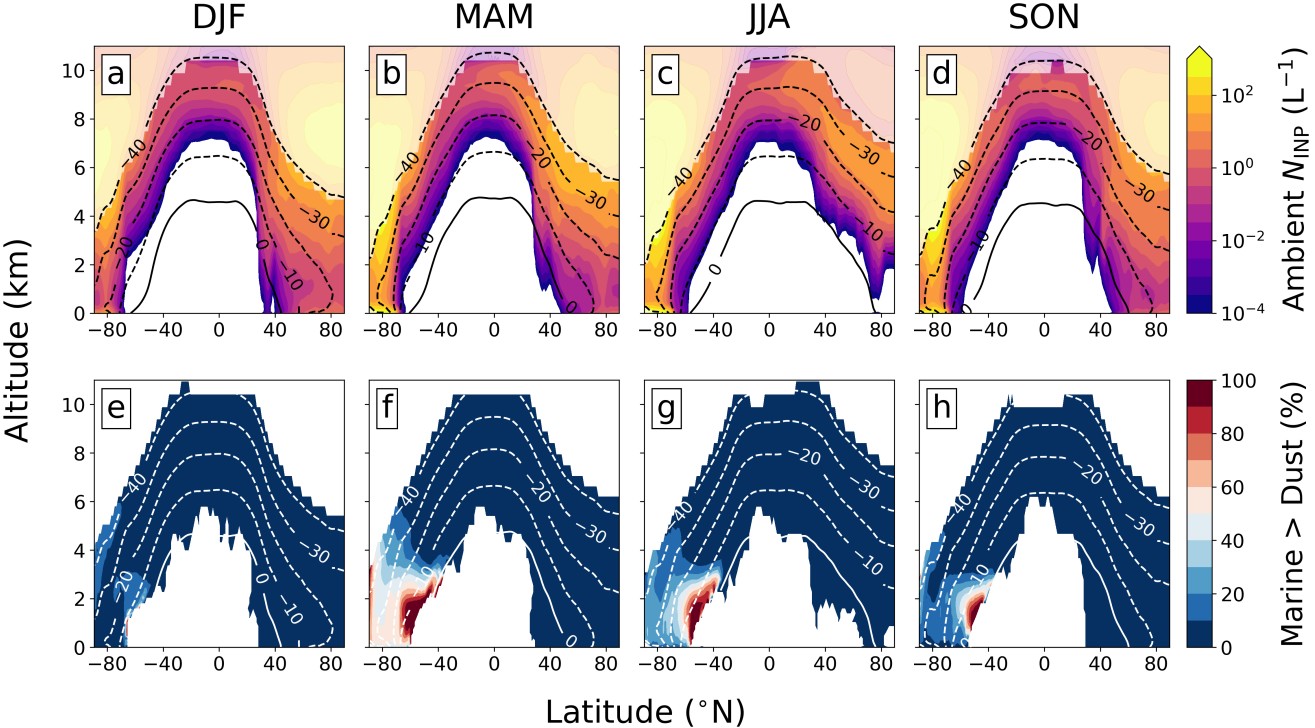

**Figure 5.** Zonal mean profiles of $N_{\text{INP}}(T_{\text{amb}})$ across the four seasons between the surface and an altitude of 11 km. Panels (a) – (d) show the total $N_{\text{INP}}(T_{\text{amb}})$ with contour lines showing the seasonal mean ambient temperature at 10 °C intervals. Data for temperatures below -40 °C are muted as homogeneous freezing is likely to dominate over heterogeneous freezing. Filled contours in panels (e) – (h) show the percentage of gridboxes (over days and longitudes) where marine sources of $N_{\text{INP}}(T_{\text{amb}})$ are larger than the contribution from the dust sources of $N_{\text{INP}}(T_{\text{amb}})$.

MAM, results in very low temperatures and high INP concentrations close to the surface. These simulations are similar to Vergara-Temprado et al. (2017), but UKESM1 simulates a much smaller contribution of marine-sourced INPs to $N_{\text{INP}}(T_{\text{amb}})$. This could be due to the different mechanisms used to represent marine INPs: in our study we use the SSA approach, whilst Vergara-Temprado et al. (2017) use the PMOA approach which predicts more INPs in Southern Ocean. This will be discussed further in Sect. 4.2.3.

## 4.2 Comparison with INP measurements

In this section we compare our INP model with a large set of ambient INP measurements. We use the relevant measurements used in Vergara-Temprado et al. (2017), as well as a selection of recent measurements not included in that study (Ardon-Dryer and Levin, 2014; Price et al., 2018; McCluskey et al., 2018b; Si et al., 2018; Irish et al., 2019; Wex et al., 2019; Si et al., 2019; Jiang et al., 2020; Tobo et al., 2020; Sanchez-Marroquin et al., 2020, 2021; Tatzelt et al., 2021; Harrison et al., 2022) (see Table



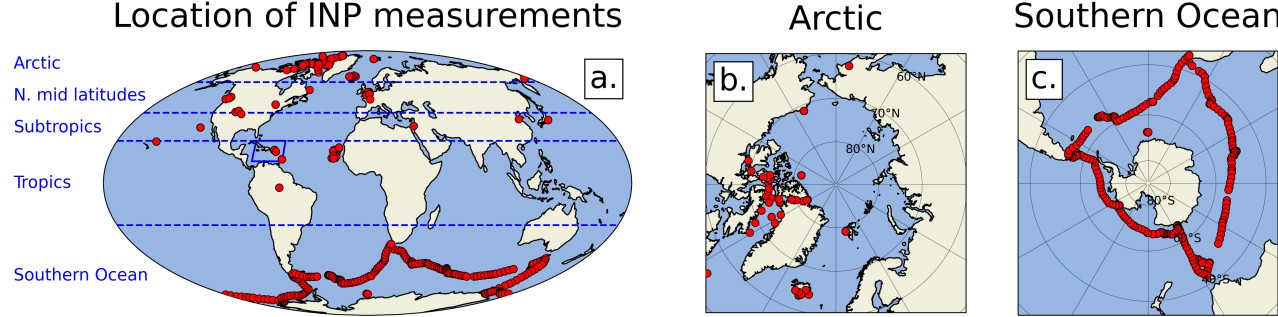

**Figure 6.** Map indicating the location of the INP measurements used for the evaluation of our model across the globe (a), the Arctic (b), and the Southern Ocean (c). Some markers represent multiple measurements taken from fixed stations across the measurement time period. The measurements took place across several different time periods and not only during the year of our simulation (2010). Note that the amount of markers in the map is not representative of the number of INP measurements. Several measurements carried out in a fixed ground based station are represented using a single marker, given the fact that the location does not change across the measuring period, as opposed to ship or aircraft based measurements. Panel (a) shows the latitudinal divisions used in this analysis, which includes the Caribbean region marked by the square box.

A1 for details; note we haven't attempted to compile all available atmospheric INP data, rather we selected a subset of INP measurements with good global coverage). Figure 6 shows the location of all the measurements used in the comparison. For further analysis we split the measurements into zonal bands for the Arctic, NH mid-latitudes, NH subtropics, tropics, and the Southern Ocean. We additionally subset the Caribbean region to highlight the skill of the model in a region downwind of the Saharan dust source.

### 4.2.1 Global evaluation

The simulated and measured $N_{\mathrm{INP}}$ are compared in Fig. 7. The INP model captures the overall spatial and temporal distribution of measured INP concentrations, which shows that the model is capable of representing the sources and transport of key INP types. The majority of the measurements are within one and a half orders of magnitude of the simulated INP concentrations, which is comparable with the spread in INP concentrations typically observed at a given location over a month (e.g., Vogel et al. (2024)). However, there is some discrepancy (Root Mean Squared Error (RMSE) of 1.81 and Mean Bias Error (MBE) of -1.37), with the model under-predicting INP concentrations, especially at lower concentrations and higher temperatures.

Figure 7b shows the ratio between the simulated and measured $N_{\mathrm{INP}}$ as a function of temperature. In general, the under-estimation of the INP concentrations is greater at higher temperatures, which explains why the bias is greatest for low INP concentrations in Figure 7a. The higher bias at low concentrations was also reported by Vergara-Temprado et al. (2017), but here we show how the bias is related to the temperature rather than the concentration. A temperature-dependent bias shows that the cause is most likely related to the assumed INP properties of the particle (i.e., INP activity versus temperature) rather than the concentration of the aerosol particles themselves, so is not related to erroneous aerosol removal rates in the model





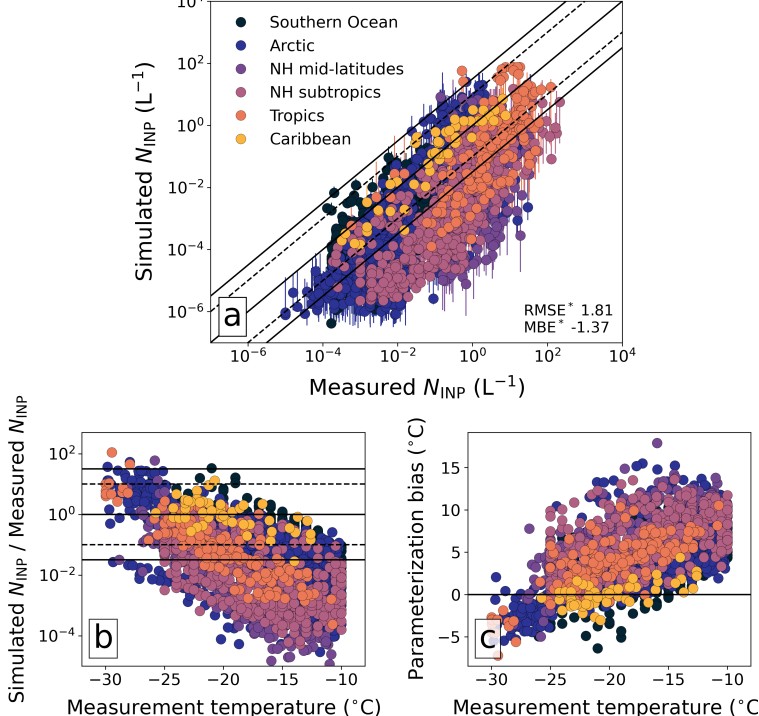

**Figure 7.** Evaluation of the INP model. (a) Simulated $N_{\mathrm{INP}}$ vs measured $N_{\mathrm{INP}}$. The simulated concentration shown is the monthly median for the corresponding month of the measurement. Bars correspond to the variability in the monthly medians throughout the year in the model. Diagonal lines correspond to a bias of zero, one order of magnitude (dashed) and one and a half orders of magnitude respectively. Values of RMSE and MBE in $\log_{10}$ space are shown in the bottom right corner. (b) Ratio between the simulated and measured $N_{\mathrm{INP}}$ versus measurement temperature. Horizontal lines correspond to a bias of zero, one order of magnitude and one and a half orders of magnitude respectively. (c) Parameterization bias in the model for each measurement.

unless such errors are also very strongly temperature dependent. Figure 7c shows how much the temperature would have to be
adjusted in the parameterization to obtain each of the same INP measurements. We define this as the temperature-dependent
"parameterization bias" and use the metric ΔPbias to characterize the temperature-dependence of the parameterization bias
for the global (or regional) dataset. ΔPbias is calculated as the difference between the mean temperature bias at temperatures
higher than -17 °C and lower than -22 °C. ΔPbias is -4.3 °C for the global dataset shown in Fig. 7c and can be attributed to
the bias at the higher end of the temperature spectrum, which is consistent with a missing source of INP that is active at higher
temperatures. One such candidate is a contribution of ice-active organic material that is known to be associated with many soils
around the world (O'Sullivan et al., 2014; Tobo et al., 2014; Steinke et al., 2016, 2020; Chen et al., 2021). Additionally, or
alternatively, bias may arise from an unrepresentative parameterization of mineral dust. We revisit these topics in Sect. 5 after
evaluating INP properties in different regional environments.



### 4.2.2    Regional evaluation

Figures 8a – f show model simulations versus measurements separated into the six regions. Here we see that the model performs better in some regions than others. Over the Southern Ocean simulated INP concentrations mostly lie within one and a half orders of magnitude, with an overall negative bias at all concentrations. In the Arctic the model performs well at high INP concentrations but is increasingly biased low towards low concentrations. The biases are similar in the NH subtropics, which primarily includes measurements from Tokyo (Tobo et al., 2020) and Tel Aviv (Ardon-Dryer and Levin, 2014). The NH

mid-latitude (western Europe and North America) measured concentrations show very little variation, with concentrations generally between $10^{-1}$ and $10^2$ L$^{-1}$. In contrast the model predicts concentrations ranging from $10^{-6}$ to $10^1$ L$^{-1}$. This represents a considerable under-prediction in our INP model's ability to represent this region. The tropics include measurements made primarily in Cape Verde and the Amazon. The Amazon measurements (mostly at higher INP concentrations) are well represented in the model, but Cape Verde concentrations are consistently under-predicted. Measurements from the Caribbean

region are well reproduced by the INP model and largely fall along the one-to-one line. This region is strongly influenced by trans-Atlantic dust from central Africa (Harrison et al., 2022), so it is surprising that we under-predict at Cape Verde but not downwind in the Caribbean. A potential reason for this is an unrepresentative K-feldspar fraction applied to Cape Verde dust, which has been reported to be much greater in Cape Verde than in the Caribbean (Glaccum and Prospero, 1980; Prospero et al., 1981). It has been reported that the K-feldspar content in Cape Verde is around 20% (Kandler et al., 2011), whereas in

Barbados values of around 1% have been reported (Harrison et al., 2022; Kandler et al., 2018) In addition, mineralogical maps show spatial variability in feldspar content (Nickovic et al., 2012), and that some regions (e.g., the source of Caribbean dust; the Sahel) have a feldspar content close to our assumed 5 % whilst other regions (e.g., the Sahara) have higher contents. This spatial variability in feldspar content in both source regions and of transported dust in the atmosphere could perhaps explain why we under-predict the INP concentration in Cape Verde. This will be explored further in Sect. 5.1.

Figures 8g – l show the temperature-dependent parameterization bias in each region. The regional analysis is consistent with the global dataset (Fig. 7c). At temperatures higher than -15 °C all regions display a positive bias (temperature applied to parameterizations is too high), whereas at lower temperatures the parameterization bias tends towards negative values. Each region exhibits this behaviour but this is more apparent in some than others. In the Caribbean there is a very weak temperature-dependent parameterization bias, whereas in the Arctic and NH mid-latitudes there is strong temperature-dependent bias.

This is consistent with the regional variations in Figs. 8a – f, suggesting that in most regions the model is missing a source of INPs active at higher temperatures (above about -15 °C). The analysis suggests relatively poorer model representation at latitudes north of 23.5 °N, which coincides with NH sources of pollution, including North America, Europe, and parts of Asia. Previous studies have found that samples of anthropogenic aerosol such as ash from combustion processes can act as INPs (Umo et al., 2015, 2019; McGraw et al., 2020). However, studies focusing on the impact of urban pollution report opposing

responses. The study of Zhao et al. (2019) report evidence of enhanced heterogeneous ice formation in deep-convective clouds under heavy pollution, whilst Adams et al. (2020) reported a negligible impact on INP concentrations following a major localised combustion event. Similarly, Chen et al. (2018) found that INP concentrations in Beijing did not correlate with





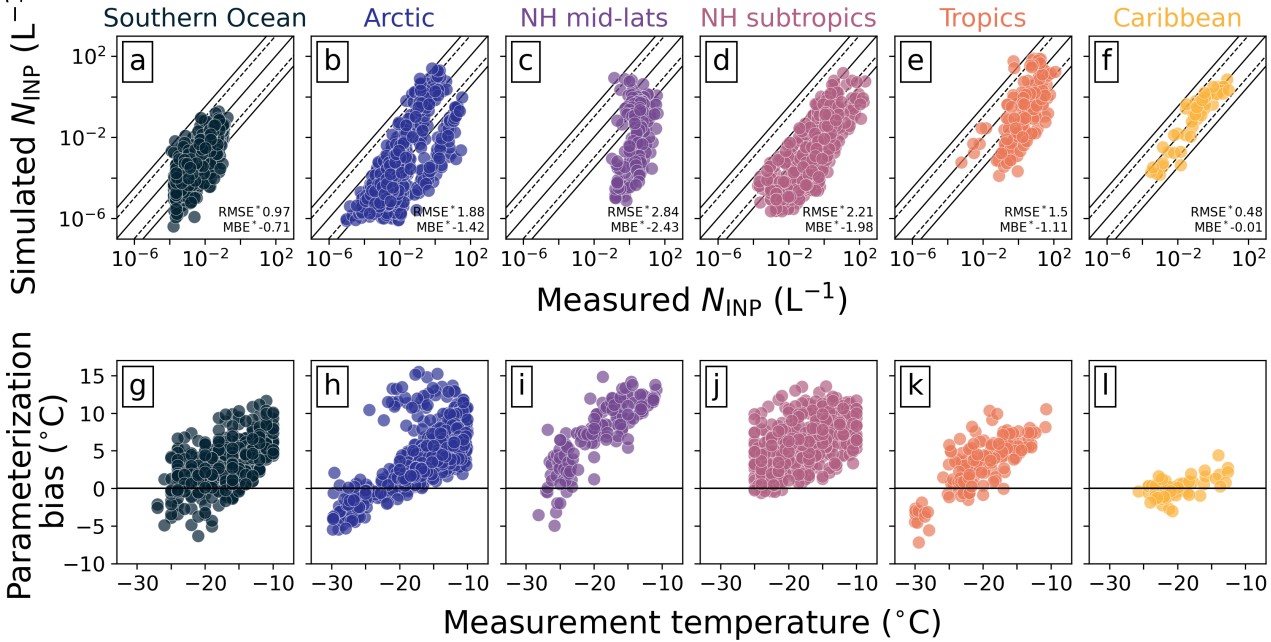

**Figure 8.** $N_{\mathrm{INP}}$ (a – f) and parameterization bias (g – l) as shown in Figs. 7a and 7c separated into the six regions of interest. The top row includes values of RMSE and MBE in $\log_{10}$ space. Diagonal lines in (g) – (l) correspond to a bias of zero, one order of magnitude (dashed) and one and a half orders of magnitude.

PM2.5 concentrations. Contemporary studies of ice nucleation by black carbon particles also suggest that black carbon plays a secondary role (at most) in the atmospheric INP population (Vergara-Temprado et al., 2018a; Schill et al., 2020). Although it is
difficult to rule out a contribution from anthropogenic aerosols, the temperature range at which soot and ash is observed to be active (generally lower than ∼-15 °C; Murray et al. (2012); Umo et al. (2015)) is not high enough to explain the temperature-dependent bias in the NH. The obvious INP class that is thought to be common in the terrestrial mid-latitudes and is active above ∼-15 °C is that of biological/biogenic INPs (Kanji et al., 2017). We return to the topic of biogenic INPs in Sect. 5.2.

### 4.2.3    PMOA versus SSA in the Southern Ocean

In this section we focus on the effect of assuming the marine-sourced INP concentrations are related to SSA or more-specifically the PMOA component of the SSA particles. We restrict the comparison to the Southern Ocean as this is the only region that we have measurements of both PMOA and SSA (Sect. 3), and is where our simulated marine-sourced INPs are most important (Fig. 4).

     Figure 9 shows the marine $N_{\mathrm{INP}}$ at -20 °C calculated using the PMOA approach (Fig. 9a) and SSA approach (Fig. 9b).
Figure 9c shows the difference in the order of magnitude between the two approaches. The PMOA approach produces between a half and two orders-of-magnitude higher $N_{\mathrm{INP}}$ than the SSA method. The difference between them is smaller in the northern





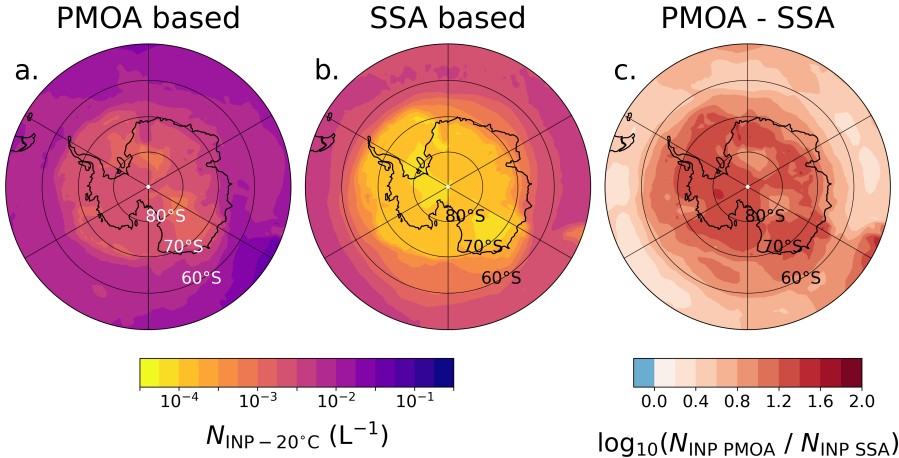

**Figure 9.** Comparison of simulated annual-mean marine-sourced INP concentrations using the two approaches (Sect. 2.3) (a) Marine $N_{\mathrm{INP}}(-20\ ^\circ\mathrm{C})$ concentration calculated using the PMOA approach at ground level. (b) Marine $N_{\mathrm{INP}}(-20\ ^\circ\mathrm{C})$ concentration calculated using the SSA approach at ground level. (c) Logarithmic difference between marine $N_{\mathrm{INP}}(-20\ ^\circ\mathrm{C})$ simulated using the PMOA and the SSA approaches (1 indicates a factor 10 ratio). The area of the calculations has been restricted to the Southern Ocean for the reasons stated in Sect. 2.3.

regions of the Southern Ocean ($50 - 60\ ^\circ\mathrm{S}$) as well as close to Antarctica. The spatial differences arise due to the underlying mechanisms of PMOA and SSA production, as well as removal processes in the aerosol microphysics scheme. PMOA emissions are dependent on surface concentrations of chlorophyll, whereas SSA emissions are wind-speed dependent. SSA is

removed from the atmosphere quicker than PMOA (Mann et al., 2010) as it is emitted into larger size modes. Differences in hygroscopy between the two species also affect the removal rate through activation as cloud condensation nuclei. This acts to relatively enhance the transport of PMOA to the Antarctic landmass (Fig. 9c). The resulting latitudinal gradient (N – S) of the two aerosol types is more negative for SSA, driving an increasing difference in $N_{\mathrm{INP}}$ between the two approaches.

To evaluate the two approaches we use measurements of $N_{\mathrm{INP}}$ from McCluskey et al. (2018a) and Tatzelt et al. (2021).

McCluskey et al. (2018a) made INP measurements during March 2016 as part of the Clouds, Aerosols, Precipitation, Radiation, and atmospherIc Composition Over the southeRN ocean (CAPRICORN) campaign, and Tatzelt et al. (2021) made INP measurements during the Antarctic Circumnavigation Expedition (ACE). ACE took place between December 2016 and March 2017, and was split into three legs (see Fig. S1 in the supporting information): Leg 1 took place from December to January south of the Indian Ocean, Leg 2 from January to February south of the Pacific Ocean, and Leg 3 from February to March south

of the Atlantic Ocean. Though INP measurements from the Southern Ocean are scarce the spatial coverage of the combined dataset provides valuable information from across the entire Southern Ocean, albeit for a single year.

Figure 10 shows that inclusion of marine-sourced INPs, from either approach, significantly enhances $N_{\mathrm{INP}}$ when compared to dust-sourced INPs alone, especially at higher temperatures. In all cases, incorporating the marine source of INPs improves the agreement of the simulated and measured $N_{\mathrm{INP}}$. For the McCluskey et al. (2018a) dataset (Fig. 10a), the SSA method





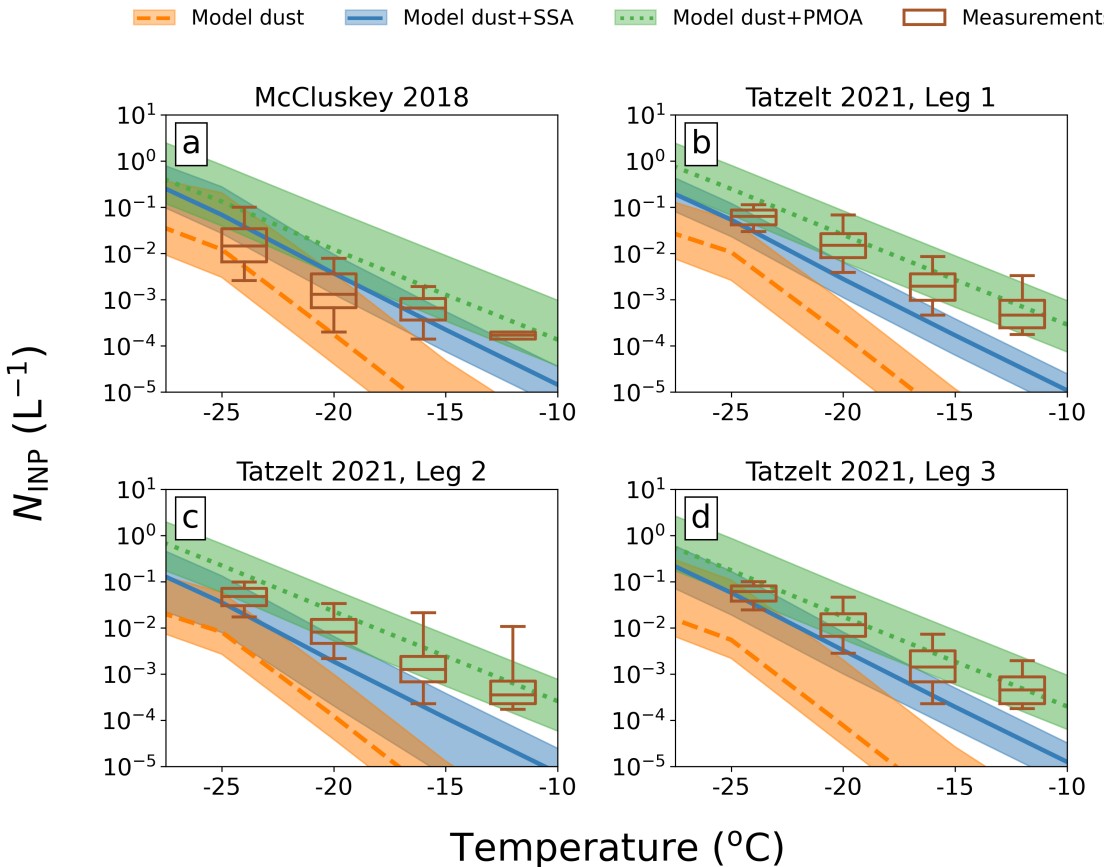

**Figure 10.** Comparison between simulated and measured $N_{\text{INP}}$ in the Southern Ocean. The panels include (a) the measurements of Mc-Cluskey et al. (2018a), and (b) – (d) the measurements reported for each of the legs of the circumnavigation by Tatzelt et al. (2021) (corresponding to the Indian, Pacific and Atlantic Oceans, see Fig. S1). The simulated $N_{\text{INP}}$, shown as lines and associated shading, is calculated for $N_{\text{dustINP}}$ (model dust), $N_{\text{dustINP}} + N_{\text{marineINP,SSA}}$ (model dust+SSA), and $N_{\text{dustINP}} + N_{\text{marineINP,PMOA}}$ (model dust+PMOA). Lines correspond to the median modelled concentration, while the shading corresponds to the 95 % interval over a time period and area comparable to the measurements. The measurements show the median, 25th and 75th percentiles of the data. The whiskers correspond to the 5th and 95th percentiles. Note that some data above -10 °C has been excluded from the analysis due to insufficient data at these higher temperatures.

produces a much better agreement to the measurements, whilst the PMOA-based method using the parameterization given by Wilson et al. (2015) tends to overestimate the measured concentrations. This is consistent with the results of McCluskey et al. (2019). However, for the three legs of the Tatzelt et al. (2021) dataset the approach that best matches the measurements is dependent on temperature. At lower temperatures (less than -20 °C) the measurements fall between the two approaches, whereas at higher temperatures, where most of the measurements are, the measurements are better reproduced by the PMOA-

based approach. Incorrectly simulated aerosol concentrations (of SSA or PMOA) would result in a systematic bias in the



simulated $N_{\text{INP}}$, therefore the temperature dependence of the bias suggests that the cause is related to the parameterizations of marine INP activity, or that the model lacks an INP aerosol species that is active at higher temperatures. We will discuss this in the following section.

The PMOA-based approach produces higher concentrations than the SSA-based approach and better reproduces the mea-
surement dataset. However, given the limited measurements in the region it is difficult to state which approach is ultimately more appropriate. The PMOA parameterization is based on sea surface microlayer samples from the Arctic and the North Atlantic (Wilson et al., 2015) and inspection of that data shows that the ice-nucleating activity (per mass of organic carbon) varies by up to a factor of 100 at any one temperature, with generally lower values found for samples in the North Altantic compared to samples from the Arctic collected in the vicinity of sea ice. Comparable measurements in the Southern Ocean are not avail-
able. The SSA-based method used in this study underestimates some of the measurements given by Tatzelt et al. (2021) but produces an INP concentration within an order of magnitude of the measurements. We conclude that the SSA method produces a valid distribution of marine $N_{\text{INP}}$ over the Southern Ocean and given it is far more simple to implement we recommend its use. It would be beneficial to repeat this analysis in other Oceans, and for other years, but this requires additional measurements that are currently lacking.

## 5    Evidence for additional INP types

The analysis in Sect. 4 suggests our INP model is unable to reproduce the temperature dependence of $N_{\text{INP}}$, with substantial bias at higher temperatures where INP concentrations are lowest. This suggests that INP active at temperatures higher than -20 °C are either missing, or currently incorrectly represented by our parameterizations. In this section we discuss potential causes of this discrepancy. Given that dust aerosol is the primary source of INP in our model (Fig. 4) we focus on this aerosol
species, though we also explore the potential for other species that are currently not simulated/represented by UKCA.

### 5.1    K-feldspar distribution and parameterizations

Several studies have shown that the ice-nucleating behaviour of mineral dust is primarily driven by the K-feldspar component (Atkinson et al., 2013; Harrison et al., 2016, 2019; Price et al., 2018; Harrison et al., 2022), which is the assumption we employ in this study. We have applied a constant value of 5 % K-feldpsar content in our simulated dust, which is characteristic
of transported dusts measured by Atkinson et al. (2013) and Harrison et al. (2019). However, we know that the K-feldspar content of soil and dust at the source is spatially variable (Nickovic et al., 2012), varying from $\sim 1 - 10\%$ (Jeong, 2024), and that larger dust particles will tend to contain relatively more K-feldspar than smaller particles (Nickovic et al., 2012). In Vergara-Temprado et al. (2017) the K-feldspar fraction of emitted dust in the model was driven by global maps of soil feldspar content, and was enhanced in the larger aerosol size modes. Using this method results in a larger K-feldspar fraction
close to dust source regions and smaller fraction in remote regions. Despite the differences, the results and conclusions of that earlier study are very similar to ours, especially in the remote regions (e.g., Southern Ocean) where marine sources of INPs out-compete dust sources for much of the annual cycle. Therefore, it is reasonable to conclude that the missing high-



temperature INPs in our model are not due to the spatial variability of K-feldspar that we neglect. However, the comparison between measurements in the Caribbean and the eastern Tropical Atlantic indicate that there is some benefit in representing

the variability in K-feldspar fraction in a similar manner to Vergara-Temprado et al. (2017).

A second source of potential model error arises from the temperature-dependent parameterization that links the dust K-feldspar content to its ice-nucleating ability. To explore this we have recalculated the global distributions of $N_{\mathrm{INP}}$ using an alternative temperature-dependence of INP activity given by Atkinson et al. (2013), shown in Fig. 11a alongside the default of Harrison et al. (2019) used in our INP model. We similarly assume a K-feldspar content of 5 %. This parameterization

results in higher $n_s$ values compared to Harrison et al. (2019) across much of the temperature range, but is consistent with the upper range of K-feldspar ice-nucleating activity from the compilation of literature data used by Harrison et al. (2019). The alternative K-feldspar parameterization leads to slightly better agreement between the model and the measurements (Fig. 11) in all regions except for the Caribbean where the RMSE increases. Table 1 shows that globally, the RMSE decreases from 1.81 to 1.55, $\Delta$Pbias has reduced from -4.3 °C to -3.1 °C, and the percentage of data with parameterization biases within a given

threshold is increased for all thresholds, except for the extreme cases where the parameterization bias exceeds 5 °C. This occurs because $n_s$ is higher than the original parameterization at temperatures below -12 °C (Fig. 11a). It is not possible to say whether the Atkinson et al. (2013) parameterization is more appropriate as some of the model-measurement uncertainty biases could be due to incorrectly simulated dust transport or size distributions, or K-feldspar variability as discussed previously. Future studies would benefit from INP, dust concentrations, and K-feldspar fraction measurements made over long time periods (months to

years), which would provide a means to more robustly evaluate simulated INP concentrations.

Other neglected mineralogical components may also play a role. Chatziparaschos et al. (2023) recently presented a modelling study where atmospheric dust was partitioned into components of K-feldspar and quartz. Although quartz has a lower ice-nucleating ability than K-feldspar (Harrison et al., 2019), it is more abundant than K-feldspar (Atkinson et al., 2013). Chatziparaschos et al. (2023) found that although K-feldspar dominated the overall contribution to dust-sourced INP concen-

trations, there were some regions and some altitudes where the quartz was more important. The greatest impact was found at low temperatures around -30 °C, therefore the addition of this component may improve our INP model at these temperatures but is unlikely to explain the missing INPs at higher temperatures. However, the active sites on quartz samples have been shown to be unstable, with some freshly milled quartz samples becoming less active with time when exposed to both liquid water and air (Harrison et al., 2019; Kumar et al., 2019). In contrast, the active sites on K-feldspars are stable with activity being main-

tained after 10 days in liquid water, and a shift of only $\sim 1$ °C after 16 Months (Harrison et al., 2016). The parameterization for quartz is for freshly ground material, which might be relevant for particles that have recently undergone the saltation process, but the aging of quartz in the atmosphere may lead to an overestimate in the contribution of quartz to the INP population.

## 5.2 Terrestrial biological material

Previous studies have shown that terrestrial biological material contributes to the INP population (Schnell and Vali, 1976;

Conen et al., 2011; Morris et al., 2014; Tobo et al., 2014; Fröhlich-Nowoisky et al., 2015; Steinke et al., 2016; Augustin-Bauditz et al., 2016; O'Sullivan et al., 2018; Steinke et al., 2020; Cornwell et al., 2023; Gong et al., 2022). This can be in the form of



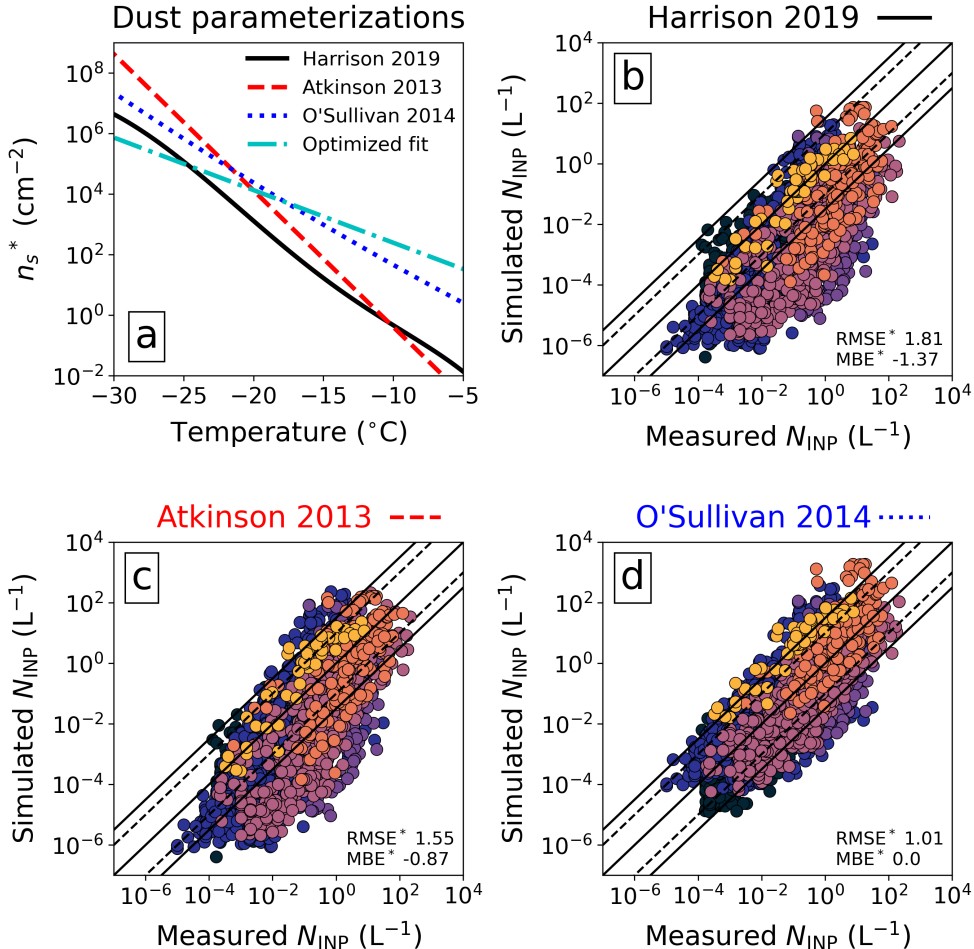

**Figure 11.** Sensitivity of modelled dust INP concentrations to parameterization of ice-nucleating activity. (a) temperature-dependent $n_s$ curves of the parameterizations for dust INPs, where $n_s$ for the Harrison et al. (2019) and Atkinson et al. (2013) curves are scaled by 5 % (accounting for dust K-feldspar content) to make all descriptions comparable. Panels (b), (c), and (d) show the simulated $N_{INP}$ vs measured $N_{INP}$ in all regions for the Harrison et al. (2019), Atkinson et al. (2013), and O'Sullivan et al. (2014) parameterizations. Diagonal lines in (b) – (d) correspond to a bias of zero, one order of magnitude and one and a half orders of magnitude respectively. The RMSE and MBE for each comparison is shown in the lower right of each panel, calculated using the $\log_{10}$ of each dataset.

primary biological aerosol particles such as fungal fragments, bacteria and pollen (O'Sullivan et al., 2015; Fröhlich-Nowoisky et al., 2015; Palmero et al., 2011; Pouleur et al., 1992; Morris et al., 2014; Steinke et al., 2020), or in the form of organic material, that ultimately results from biological activity, such as macromolecules or other detritus attached to agricultural soil

particles (O'Sullivan et al., 2015; Conen et al., 2011; O'Sullivan et al., 2016; Hill et al., 2016). Recent measurements in the Arctic suggest that dust emissions from high-latitude river sediment sources have similar activity to agricultural soils and also contain an important ice active organic component (Barr et al., 2023; Meinander et al., 2022; Tobo et al., 2019).



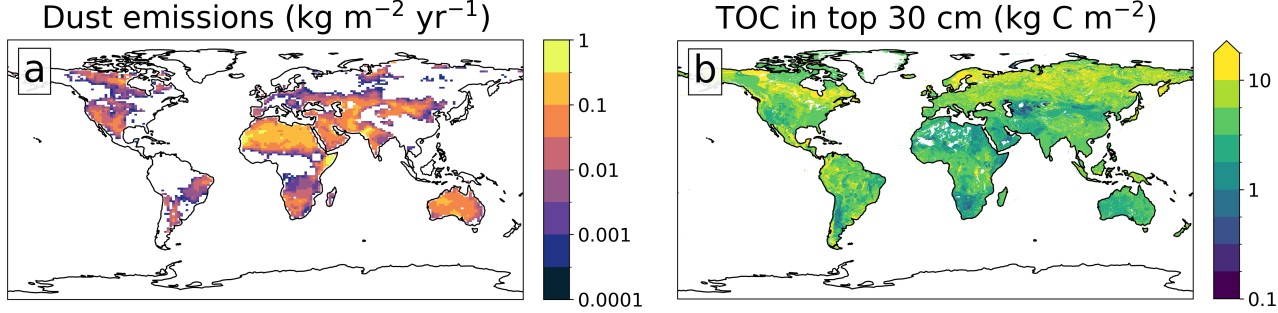

**Figure 12.** Annual mean simulated dust emissions (a) and total organic carbon (TOC) content in the topsoil (0 – 30 cm) layer (b) taken from FAO et al. (2012).

Figure 12 shows that the total organic carbon (TOC) content of the topsoil layer can vary by around one order-of-magnitude across relatively small spatial scales. Some dust-producing regions such as the Sahara desert have typically low TOC contents (<1 kg C m$^{-2}$), whereas other dust emission regions such as parts of North America and central Russia have relatively high TOC content (>10 kg C m$^{-2}$). We might expect the contribution of ice-active organic material to dust INPs to be larger close to regions where there are periodically conditions commensurate with biological activity (i.e., liquid water), such as agricultural or seasonally fertile areas as well as river flood plains. In contrast, dust from the most arid deserts, such as much of the Sahara, has a very low organic carbon content (Minasny et al., 2017, 2014; Price et al., 2012), and measurements indicate that dusts from these arid sources are controlled by their mineral content rather than any trace of organic material (Harrison et al., 2022) (see Fig. S3).

Although some studies have attempted to include pollen grains, bacteria and fungal spores in global models (Hoose et al., 2010; Spracklen and Heald, 2014; Cornwell et al., 2023), the lack of quantitative understanding of factors that determine fluxes of biological particles and of their ice-nucleating behaviour limits our capacity to represent all biological sources of INPs. Ice-nucleating proteins from soil fungus are known to bind to mineral dust particles and retain the ice-nucleating ability (O'Sullivan et al., 2016), whilst the membrane-bound proteins from bacteria can still nucleate ice after the cells are no longer viable and are fragmented (O'Sullivan et al., 2015; Murray et al., 2022; Bieber and Borduas-Dedekind, 2023). Additionally, pollen ice-nucleating macromolecules can be readily separated from the original pollen grains (Pummer et al., 2012; Murray et al., 2022). Hence, it has been suggested that these biological macromolecules or fragments of ice-nucleating biological entities might serve as a much more numerous group of INPs in the atmosphere than whole bacterial cells, pollen grains or fungal spore (Rangel-Alvarado et al., 2015; O'Sullivan et al., 2015). Furthermore, the work that has been done on the global contribution of pollen, bacteria and fungal spores to INP concentrations indicates that these species contribute very few INPs because particle concentrations in the atmosphere are much lower than concentrations of dust or SSA Spracklen and Heald (2014); Hoose et al. (2010).

Here we explore the potential effect of organic material attached to dust particles. We refer to soil dusts that contain organic material that results from biological activity as *fertile soils* and use *desert soils* to refer to the soils from arid regions containing



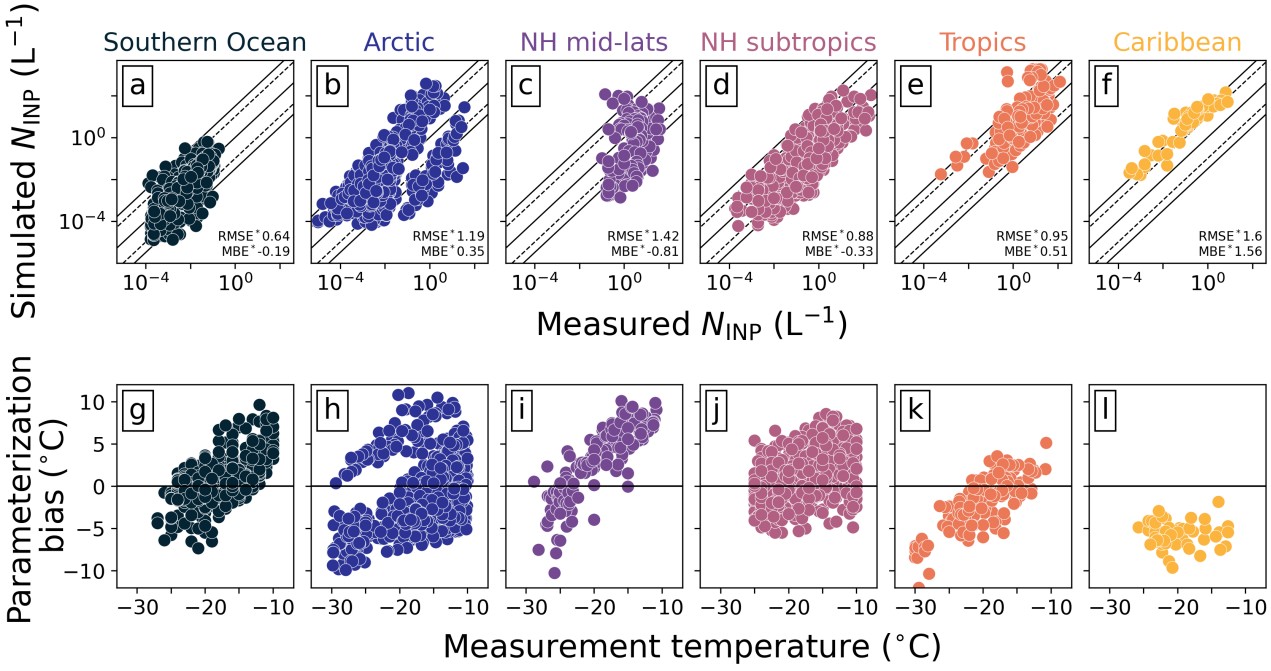

**Figure 13.** Evaluation of the INP model using the fertile soil parameterization of O'Sullivan et al. (2014) to represent dust INPs. (a) – (f) regional comparison between the simulated and measured concentrations and (g) – (l) parameterization bias as a function of measurement temperature in each region. Simulated INP concentrations shown in (a) – (f) are monthly medians for the corresponding month of the measurement. Diagonal lines in (a) – (f) correspond to a bias of zero, one order of magnitude (dashed) and one and a half orders of magnitude.

very little organic matter. Fertile soils will likely produce mineral dust aerosol with more organic material associated with it than desert soils. In reality the biological content will vary between regions, e.g., Fig. 12b. Even soils in some semi-arid regions can contain organic matter and pose as a viable INP source (Schnell, 1974a, b), though airborne dust from North

Africa was found to contain no observable biological ice-nucleating components (Harrison et al., 2022). Although the global database of TOC in Fig. 12b may suggest a potential link between dust emissions and biological content, at present there are insufficient INP measurements to define the variability of activity we might expect due to the biological activity of fertile and desert soils. Therefore, as a simple first step, and a reasonable approximation, we represent the ice-nucleating ability of all mineral dust as fertile soil according to the parameterization of O'Sullivan et al. (2014). The soils from the UK that O'Sullivan

et al. (2014) analysed are consistent with the ice-nucleating activity of Argentinian soil dust (DeMott et al., 2018), Wyoming soil dust (Tobo et al., 2014), and high-latitude proglacial dust from the Copper River Valley (Barr et al., 2023). A comparison of $n_s$ (Fig. 11a) shows that the fertile soil parameterization is two orders-of-magnitude more active than that of K-feldspar (desert soils) at temperatures higher than -12 °C, but is within a factor of 10 at temperatures below -20 °C. By representing





**Table 1.** Table showing: (top half) the RMSE and MBE between measurements and simulations using different dust INP parameterizations from the literature (Harrison et al., 2019; Atkinson et al., 2013; O'Sullivan et al., 2014) and an optimized parameterization; (middle) temperature-dependence of the bias expressed using the metric ΔPbias; and (bottom half) percentage of data within parameterization-bias thresholds.

| | Harrison 2019 | Atkinson 2013 | O'Sullivan 2014 | Optimized fit |
|---|---|---|---|---|
| **RMSE and MBE (in parantheses) of measurements and model data (in $\log_{10}$ space)** | | | | |
| Global | 1.81 (-1.4) | 1.55 (-0.9) | 1.01 (-0.0) | 0.9 (-0.0) |
| Arctic | 1.88 (-1.4) | 1.71 (-0.9) | 1.19 (0.4) | 1.1 (0.4) |
| NH mid-latitudes | 2.84 (-2.4) | 2.34 (-1.6) | 1.42 (-0.8) | 1.3 (-1.0) |
| NH subtropics | 2.21 (-2.0) | 1.78 (-1.3) | 0.88 (-0.3) | 0.8 (-0.3) |
| Tropics | 1.5 (-1.1) | 1.12 (-0.4) | 0.95 (0.5) | 0.6 (0.2) |
| Caribbean | 0.48 (-0.0) | 0.93 (0.7) | 1.56 (1.6) | 1.4 (1.3) |
| Southern Ocean | 0.97 (-0.7) | 0.87 (-0.6) | 0.64 (-0.2) | 0.5 (0.4) |
| **Temperature dependence of parameterization bias ($\Delta\mathrm{Pbias} = \bar{\mathrm{Pbias}}_{>-17°\mathrm{C}} - \bar{\mathrm{Pbias}}_{<-21°\mathrm{C}}$)** | | | | |
| | -4.3 °C | -3.1 °C | -3.2 °C | -0.6 °C |
| **Percentage of data within parameterization bias threshold ($\pm$)** | | | | |
| $\leq 1°$C | 12% | 17% | 25% | 19% |
| $\leq 2°$C | 25% | 32% | 45% | 37% |
| $\leq 3°$C | 37% | 46% | 59% | 51% |
| $\leq 4°$C | 50% | 57% | 71% | 61% |
| $\leq 5°$C | 61% | 66% | 81% | 70% |
| $\leq 10°$C | 93% | 89% | 100% | 96% |

the ice-nucleating ability of all mineral dust with the ice-nucleating ability of fertile soil we expect that we will over-estimate
the INP concentration in some locations (such as the Caribbean where dust was found to contain no appreciable organic ice
nucleating material), but this test will tell us if the organic material associated with fertile soils is a potential explanation for
low-biased INP concentrations elsewhere in our model.

Figure 11d compares the model and measurements assuming a fertile soil parameterization for all dust. A regional comparison of $N_{\mathrm{INP}}$ and parameterization bias is shown in Fig. 13. A fertile soil assumption considerably improves model-
measurement agreement and removes a substantial fraction of the temperature-dependent bias in $N_{\mathrm{INP}}$. The ΔPbias has
decreased from -4.3 °C to -3.2 °C, and global RMSE has reduced from 1.81 to 1.01 (MBE is now 0.0), contributed by improvements in all regions except for the Caribbean (Table 1). Greatest improvement occurs in the NH subtropics and mid-latitudes,
where the RMSE falls from 2.2 and 2.8 to 0.9 and 1.4. In contrast, the RMSE in the Caribbean has increased from 0.5 to 1.6.
We also see an improved representation in the Arctic where RMSE has decreased from 1.88 to 1.19. This is consistent with the



simulations of Kawai et al. (2023) who found that summertime (JJA) Arctic INP measurements were best reproduced using a parameterization (Tobo et al., 2019) that accounted for measured organic content associated with locally-sourced surface dust.

These fertile soil simulations suggest that the organic component in fertile soil makes an important contribution to INP activity at temperatures higher than -20 °C where INP concentrations are usually low. The results also suggest that INPs around the world could be explained by a mixture of more- or less-fertile dust particles sourced from different regions. In

the Caribbean, a fully fertile soil parameterization consistently overestimates measured INP concentrations (Fig. 13f), while the desert soil assumption (Table 1) produces realistic INP concentrations. This is a region heavily influenced by the major dust source regions of central and northern Africa where dust is mostly abiotic, as demonstrated by the heat tests performed on Caribbean samples (Harrison et al., 2022) and the low TOC values at the surface (<1 kg C m$^{-2}$; Fig. 12b). Hence, the fact that INP concentrations in the Caribbean are better represented by the desert soil parameterization is consistent with

the measurements and known sources of dust. In the Arctic, the fertile soil parameterization produces INP concentrations with mostly zero bias relative to the measurements (Fig. 13h) at temperatures higher than -15 °C, but the model is generally negatively biased below -20 °C. This suggests that the Arctic is influenced by dust sourced from both fertile and desert soils. There is good evidence that dust from the low latitudes (e.g., African continent) is transported to the high latitudes alongside dust from local sources (Shi et al., 2022), the latter of which is thought to contain a high fraction of biological material (Tobo

et al., 2019; Kawai et al., 2023). Therefore we hypothesise that representing the spatial variability of K-feldspar in desert soils and organic material in fertile soils will improve the ability of our model to simulate measured INP concentrations.

### 5.3 High-latitude dust

A third source of uncertainty we explore is the regional pattern of dust emissions. Much of the focus of global modelling centres is to better represent the size distributions and fluxes of dust from the major dust sources, such as those in Africa, the

Middle East, and Asia. Historically, less attention is given to the weaker dust source regions, which include the high latitudes. However, these regions have been identified as substantial sources of local dust (Bullard et al., 2016; Arnalds et al., 2016; Dagsson-Waldhauserova et al., 2019; Shi et al., 2022; Meinander et al., 2022) and measurements of these dust particles show high INP activity (Sanchez-Marroquin et al., 2020; Barr et al., 2023; Tobo et al., 2019; Xi et al., 2022).

Simulating high-latitude dust (HLD) is challenging given that its emission mostly occurs in glaciofluvial valleys and is

strongly affected by small-scale katabatic winds (Bullard et al., 2016; Prospero et al., 2012). The driving processes tend to occur at around 10 km scales, which are much smaller than the resolution of global models, and thus rely on sub-gridscale parameterizations. Despite the challenges, some studies have simulated HLD on global and regional scales (Leadbetter et al., 2012; Groot Zwaaftink et al., 2016, 2017; Tobo et al., 2019; Hamilton et al., 2019; Mingari et al., 2020; Sanchez-Marroquin et al., 2020; Shi et al., 2022). To explore the potential impact of HLD we use UM simulations that were performed by Walters

et al. (2019) using the atmosphere model identical to ours, but with additional coupling of a land-surface model. In the model description paper the authors introduce the Met Office UM Global Atmosphere and Global Land configuration (GA7.1 and GL7.0). As part of this study the authors ran simulations of the atmosphere coupled to the Joint UK Land Environment Simulator (JULES) at different resolutions including N96 (135 km resolution; as used in our INP model), N216 (60 km



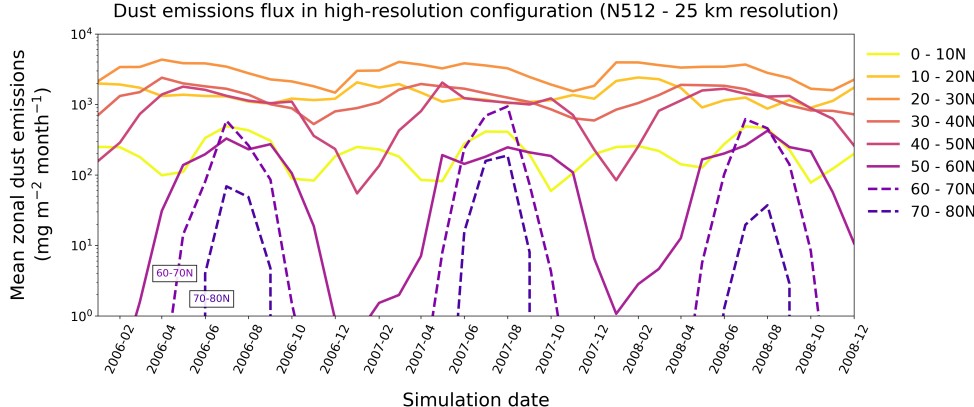

**Figure 14.** Time series (Jan 2006 – December 2008) of zonal mean dust emission fluxes in the NH from the high-resolution configuration (25 km) of UKESM simulations presented by Walters et al. (2019). Each colored line shows the mean for a 10 degree latitude band from the equator to 80 °N. Latitudes south of 60 °N are shown as solid lines, and latitudes north (the Arctic) are shown as dashed lines.

resolution), and N512 (25 km resolution). Although these are all unable to explicitly resolve turbulent motion we can use them

to gain a perspective on the resolution dependence of dust emissions in our model. For the following analysis we will focus on the Arctic.

   Figure 14 shows NH zonal mean dust emissions fluxes in the N512 resolution simulations. Peak emissions for latitudes north of 30 °N occur between April and October. In the Arctic region (north of 60 °N) simulated dust emissions are negligible (below 1 mg m$^{-2}$ month$^{-1}$) outside of these months, and peak in July and August. Measurements from HLD sources (Bullard et al.,

2016; Bullard and Mockford, 2018; Prospero et al., 2012) are consistent with the time period for simulated Arctic emissions (April – October) but suggest the month of peak emissions may occur earlier or later than July. This suggests even at the finest model resolution tested here UKESM may be unable to represent HLD emissions of dust from sub-gridscale features such as valleys. Given that simulated emissions peak in July, we will focus on this month to investigate the role of model resolution dependence.

The July-mean dust emission flux north of 60 °N is shown for the three resolutions in Figs. 15a – c. We also show two of the primary drivers (see Sect. 4) of dust emissions in the model (Woodward et al., 2022): soil moisture content (Figs. 15d – f, shown for the top soil layer) and the dust friction velocity (Figs. 15g – i). At the coarsest resolution (135 km) dust emissions occur predominantly over North America and Central Russia. As the resolution increases there are more numerous source regions, often with greater magnitudes. These new sources include coastal regions surrounding Greenland, Iceland, and the Arctic

Archipelago, which likely correspond to small-scale glaciofluvial sites. Much of northern Scandinavia becomes a widespread source of dust in the finest resolution (25 km). Table 2 provides information on the July mean dust emissions and variability for the Arctic region (north of 60 °N). As resolution increases, so does the annual mean flux and variability. Both the magnitude and variability (standard deviation) of the emissions increase three-fold from the coarsest to finest resolution. This demonstrates that the overall emissions of Arctic HLD and the likelihood of extreme cases (in the form of relatively enhanced plumes of





**Figure 15.** Dust emissions and associated surface properties during July 2008 in the UM GA7.1 GL7.0 simulations by Walters et al. (2019) at three resolutions over the Arctic region > 60 °N. Monthly mean values of (a) – (c) dust emission fluxes, (d) – (f) soil moisture content, and (g) – (i) dust friction velocity. The configurations (N96, N216 and N512) correspond to horizontal resolutions of approximately 135, 60 and 25 km.

dust) are sensitive to the model resolution. This result may be specific to the model we use and its representation of dust emission processes. As an example, Shi et al. (2022) used a global model with 100 km horizontal resolution but were capable of simulating Arctic HLD emissions using a physically-based dust emission model following Kok et al. (2014). Emissions in



**Table 2.** Table showing the annual mean dust emissions flux and standard deviation from inside the Arctic (north of 60 °N) for three different resolution configurations of the UM-UKCA coupled to JULES. The figure in the parentheses is the typical horizontal grid scale.

| | July mean HLD emissions (g m$^{-2}$ yr$^{-1}$) | Standard deviation of emissions (g m$^{-2}$ yr$^{-1}$) |
|---|---|---|
| N96 (135 km) | 0.9 | 7.1 |
| N216 (60 km) | 1.8 | 18.7 |
| N512 (25 km) | 2.6 | 22.6 |

the Kok et al. (2014) dust model are dependent on wind friction velocity and the soil's threshold friction velocity, as in our model, but also include a term that accounts for increased emissions through saltation bombardment as the soil erodibility
increases. Additionally, the model is primarily dependent on variables that are properties of the large-scale environment (soil and horizontal winds), which may help to suppress resolution-dependence.

We explore the drivers of resolution dependence by focusing on the soil moisture content (Figs. 15d – f) and dust friction velocity (Figs. 15g – i). The soil moisture content is one of the key factors for dust uplift potential, with drier soil more readily emitted. Higher friction velocity also promotes more efficient dust uplift. The spatial distribution of soil moisture is generally
similar across the scales, but the higher-resolution simulations are increasingly capable of resolving small-scale features. These include glacial valleys (visible throughout Russia and Northern Asia) or sharp gradients that are smoothed out in the coarser scale (such as around the coastlines of Iceland and Greenland). Some regions have also become noticeably wetter or drier as resolution is increased, including the Alaskan North Slope and Scandinavia. The dust friction velocity displays similar sensitivity. The finer resolution permits sharper gradients and therefore more numerous extreme cases of high friction velocities.
There is a particular sensitivity over the Arctic Archipelago, which drives some of the enhanced dust emissions. The reason behind these changes is beyond the scope of this paper, but provides additional evidence that model resolution is important for representing Arctic HLD emissions. We hypothesise that an increase in model resolution beyond that investigated here would be required in order to reduce the discrepancy between the annual cycle of HLD emissions simulated by UKESM1 (Fig. 14) and as suggested by measurements (Prospero et al., 2012; Bullard et al., 2016; Bullard and Mockford, 2018). However,
measurements of HLD are scarce, which makes evaluation difficult and provides a degree of uncertainty as to how much HLD our models should in fact simulate. This uncertainty can be observed in Shi et al. (2022) who find that although their model simulated a peak in summertime Arctic HLD at the Alert station (82.5 °N, 20.3 °W), this was not replicated in the measurements (Sirois and Barrie, 1999), which displayed either no seasonal cycle, or a summertime minimum.

This section demonstrates that Arctic HLD emissions are sensitive to the model resolution. This may also hold true in the
Southern Hemisphere, especially around the coastlines of Antarctica which are known to be potential sites for dust emissions (Asmi et al., 2018; Bory et al., 2010; Meinander et al., 2022; Kavan et al., 2017, 2018). Using a higher resolution in our simulations would likely increase the INP concentrations in our model, but this would occur primarily in the high-latitudes and at all freezing temperatures. Therefore, weak HLD emissions do not explain the missing high-temperature INPs in our simulations. However, as the soil properties play a primary role in driving dust emissions, we recognise that using an interactive





land surface model would be a beneficial addition to our study, and therefore recommend this for future studies focusing on
HLD. Finally, there is a clear need for additional information on dust emissions from high-latitude environments, especially
given that future trends suggest a shift towards more summertime melting and the appearance of either seasonal or permanent
regions of bare soil that can produce dust Meinander et al. (2022).

### 5.4   Optimized dust representation

Replacing the K-feldspar dust parameterization with a fertile soil assumption reduces the RMSE and some of the temperature-
dependent $\Delta$Pbias (Table 1), but there remains a considerable parameterization bias towards the higher end of the temperature
spectrum (Fig. 16). This suggests the INP spectrum has an even weaker temperature-dependence than the fertile soils descrip-
tion from O'Sullivan et al. (2014). An alternative parameterization for dust INPs was produced to give the lowest combination
of RMSE and $\Delta$Pbias when evaluated against the measurements. Figure 11a shows the optimized parameterization, which is

described by $n_s = \exp(-0.4T + 110.777)$, where $T$ is in Kelvin, and applicable for temperatures between -10 °C and -30 °C.
Figure 16 shows that the $\Delta$Pbias has been substantially reduced from -3.2 °C (for the O'Sullivan et al. (2014) parameterization)
to -0.6 °C, and the RMSE has reduced from 1.01 to 0.9. The optimized parameterization of $n_s$ has a shallower temperature-
dependence than the other descriptions, with a value of -0.175 °C$^{-1}$ compared to -0.35 °C$^{-1}$ and -0.28 °C$^{-1}$ for the Harrison
et al. (2019) and O′Sullivan et al. (2015) parameterizations. This temperature-dependence is consistent with studies that have

measured the ice-nucleating activity of soil and dust samples, but shallower than most known INP species. Soil samples taken
from four different regions of the world in Steinke et al. (2016) exhibit an $n_s$ gradient of -0.15 °C$^{-1}$, and dust samples from
East Asian dust events in Chen et al. (2021) exhibit an $n_s$ gradient around -0.19 °C$^{-1}$. This suggests the material causing the
shallow gradient is associated with the dust aerosol but requires further work to establish whether this is caused by organic
material or another INP species. Figure 16 shows that the median parameterization bias is now close to 0 °C for measure-

ment temperatures higher than -26 °C but there remains a negative bias below this temperature. This supports our hypothesis
that the mineral component, which exhibits a steeper temperature-dependence (Fig. 11a), is important at lower temperatures,
whereas the organic component (or material causing the shallow gradient) is important at higher temperatures. We note that
the optimized fit is a simple exponential description that, if extrapolated to 0 °C, would be at odds with classical nucleation
theory (Kashchiev, 2000). Measurements, such as those of O'Sullivan et al. (2014), demonstrate that the ice nucleating activity
of INP types often change with temperature and rapidly decrease as the temperature approaches 0 °C. We therefore anticipate

the gradient of the optimized representation, and associated ice-nucleating material, to not stay constant across the temperature
spectrum.

## 6   Conclusions

In this study we present the first global simulations of ice-nucleating particles (INPs) using the atmosphere configuration of

the UK Earth System Model (UKESM). The model is driven by two primary sources of INPs: mineral dust aerosol assuming
that its ice-nucleating ability is determined by the K-Feldspar content (Harrison et al., 2019) and marine sources associated



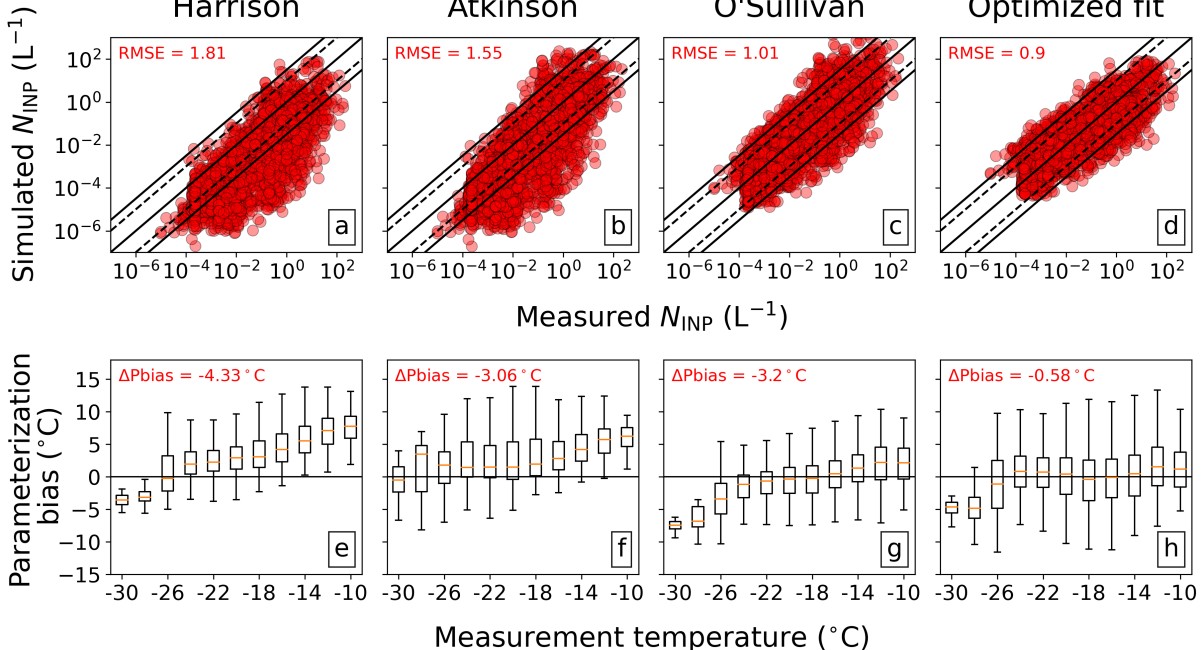

**Figure 16.** Evaluation of the INP model using four representations of simulated dust INPs including three parameterizations of $n_s$ from literature (Harrison et al., 2019; Atkinson et al., 2013; O'Sullivan et al., 2014) and the optimized parameterization. (a) – (d): Comparison between simulated and measured INP concentrations with the RMSE for each dataset shown in the upper left corner. (e) – (h): Distribution of parameterization bias (in the form of boxplots) as a function of temperature for each representation. The $\Delta$Pbias value (measure of temperature-dependence) for each representation is shown in the upper left corner of (e) – (h).

with sea spray aerosol (SSA) (McCluskey et al., 2018a) or the primary marine organic aerosol (PMOA) component of the SSA (Wilson et al., 2015). Multi-decadal simulations were run in order to quantify a mean annual cycle.

The spatial distribution of INP concentrations in UKESM is dominated by the major dust source regions, whereas marine-sourced INPs are confined primarily to the oceanic regions and have less spatial variability. In the Northern Hemisphere (NH), dust is the dominant INP in all regions, with marine-sourced INPs playing a negligible role even over the North Atlantic, North Pacific and Arctic Oceans. The dominance of dust in the Arctic region, albeit with a local supply of marine-sourced INPs, highlights the important role of long-range dust transport from the low-latitude dust source regions to the NH high-latitudes. In contrast to the NH, marine-sourced INPs are the dominant source of INPs over large parts of the Southern Hemisphere

(SH), particularly over the Southern Ocean where where only 10 – 20% of annual mean INP concentrations are dust. Over the Antarctic landmass, dust supplies about 60 – 80% of INPs, which is caused by transport of dust from non-local sources (given that we have no local sources simulated in the model). This is compatible with trajectory analysis from Neff and Bertler (2015), who found that Patagonia and New Zealand were important suppliers of dust to Antarctica, and also with McConnell et al. (2007) who have associated the desertification of Patagonia with a doubling of dust deposition to Antarctica in the



twentieth century. However, other studies have identified local dust sources (Asmi et al., 2018; Bory et al., 2010) that also may play a role, especially during Austral summer melt periods.

A comparison of simulated $N_{\mathrm{INP}}$ against a large global dataset of measurements demonstrates that our INP model is able to represent the general spatial and temporal distribution of INP, which varies by roughly four orders of magnitude (calculated at an altitude of 500 m and -20 °C). However, the model tends to underestimate measured INP concentrations particularly when concentrations are low, which we show is caused by under prediction at higher temperatures. As an average across all the measurements, the bias is equivalent to a 10 °C shift in the INP spectrum. Such a temperature-dependent bias occurs in all regions except for the Caribbean, which is downwind of the central Africa mineral dust sources. The tropics, NH subtropics and NH mid-latitudes have a particularly strong temperature-dependent parameterization bias as well as an under-prediction of most measurements. The temperature dependence of the bias suggests that it cannot be due to incorrectly simulated dust transport or dust size distributions, which would cause a consistent bias across the whole temperature range.

The INP representation and temperature-dependent model-measurement bias is improved when we assume that organic material is attached to the dust particles. The default ice-nucleation parameterization based on the K-Feldspar content of dust (Harrison et al., 2019) was replaced with a parameterization based on agricultural soil samples (O'Sullivan et al., 2014) that accounts for an additional organic component. We refer to this class as *fertile soil* and use *desert soil* to refer to soils containing very little organic matter whose ice-nucleating ability is dependent on the mineralogical component. The fertile soil parameterization enhanced the ice-nucleating activity of dust particularly at high temperatures (greater than -15 °C), reducing some of the temperature-dependent bias (ΔPbias) from -4.3 °C to -3.2 °C and decreasing the RMSE of the global dataset from 1.81 to 1.01. Most improvement was found in the NH mid-latitudes and NH subtropics where the model was originally poorest. However, agreement with the Caribbean measurements is worsened, and is instead best represented assuming no organic component. This demonstrates that not all dust sources can be assumed to be represented as fertile soils. We hypothesise that there is likely spatial variability in the ice-active organic component of fertile soil, that may be related to factors such as rainfall, soil organic carbon, or soil biodiversity. We aim to explore this in future studies.

An optimized parameterization of $n_s$ for dust INPs suggests the best representation exhibits a shallower temperature-dependent parameterization than current descriptions. The optimized parameterization substantially reduced ΔPbias from -3.2 °C for the O'Sullivan et al. (2014) fertile soils parameterization to -0.6 °C, whilst reducing the RMSE for INP concentrations from 1.01 to 0.9. This was achieved primarily due to the weaker temperature-dependence of $n_s$, which has a logarithmic slope of -0.175°C$^{-1}$, compared to -0.35°C$^{-1}$ and -0.28°C$^{-1}$ for the Harrison et al. (2019) and O'Sullivan et al. (2014) parameterizations. Though such gradients have been observed in laboratory studies (Steinke et al., 2016; Chen et al., 2021) further work is required to identify the aerosol material causing this, and whether it is associated with the dust or is a distinct INP species that needs to be represented.

Arctic INPs are particularly important for mixed-phase clouds in cold air outbreaks, and it has been hypothesised that local sources may be important (Murray et al., 2021). We therefore investigated the model's ability to represent such sources. A comparison of simulated Arctic dust emissions using three model configurations with resolutions of 135 km (this study), 60 km, and 25 km, demonstrated that summertime high-latitude dust (HLD) emissions and their variability are considerably





dependent on model resolution. Mean emissions and their variability increase three-fold from the coarsest to the finest resolu-
tion. This suggests our INP model is missing some sources of HLD, and may contribute to the misrepresentation of Arctic INP
measurements. We recommend an exploration of alternative methods to reproduce HLD emissions in the UKESM alongside
much-needed measurements from around the Arctic in order to robustly evaluate and ground the models.

Over the Southern Ocean, we found that the simulated INP concentration is sensitive to the method used to represent the ice-
nucleating ability of organics-enriched SSA. A parameterization using the PMOA content of the SSA (Vergara-Temprado et al.,
2017; Burrows et al., 2013) produces more INPs than a simpler parameterization based on the SSA surface area (McCluskey
et al., 2018b). The difference between the two approaches increases with proximity to the Antarctic landmass, which can be
linked to differences in the emission processes of PMOA and SSA. A comparison of the two methods with INP measurements
from the Southern Ocean shows that at the lowest temperatures ($\sim$ -25 °C) the range in measured $N_{\text{INP}}$ falls across both
approaches, whereas at higher temperatures the PMOA approach is more representative. The SSA method is technically easier
to represent in the model as PMOA is not regularly included as a prognostic aerosol species in aerosol microphysics models
(including UKCA). Therefore, although the PMOA method is likely the more appropriate representation we conclude that the
SSA method is a reasonable pragmatic approximation for marine-sourced INPs. We note that PMOA emissions are likely more
sensitive to future climate and/or inter-annual variability in oceanic productivity than the SSA.

Our results suggest several directions for the the improvement of global INP models. There is a clear need for evaluations to
be made on longer timescales that can capture diurnal cycles, synoptic scale events, seasonal cycles, and inter-annual variability.
This will provide a robust test of whether models can capture the key transport pathways between source and receptor regions.
The evaluations will additionally enable better separation of INPs into particular particle compositions emitted from particular
locations, which is needed to build a model that represents sources, transport and removal of INPs. For this to be possible
we require long-term INP measurements that span these timescales. Several recent campaigns in the Arctic (Sze et al., 2023;
Creamean et al., 2022) and those using the Portable Ice Nucleation Experiment (PINE) cloud chamber instruments (Möhler
et al., 2021; Brasseur et al., 2022; Adams et al., 2021; Wilbourn et al., 2023; Vogel et al., 2024) are capable of making these
measurements. In future work we aim to repeat this evaluation and focus on these longer records, and we encourage continued
effort in extending these types of measurements. The analysis of simulated INP concentrations at ambient conditions in Fig.
5 highlights that INPs are predominantly active within the lowest 5 km of the atmosphere north of 40 °N and south of 40 °S.
Therefore we recommend in-situ INP measurements are targeted within these latitudes to gain a representative dataset for
evaluation. This should be complemented by building on our existing knowledge of INP activity from surface samples at
important source regions, such as Africa, the Middle East, and Australia, and HLD sources that may play a disproportionately
important role. Finally, if dust emissions are associated with fertile soils it is likely that marine-sourced INP concentrations
in the Southern Hemisphere will become less important, to an extent that it may be reasonable to assume global INP activity
can be represented by dust alone. This would simplify the requirements for adding interactive INPs into climate models. This
hypothesis would require additional measurements in the SH, especially on longer timescales that can capture the seasonal
cycles of dust, SSA, and PMOA and help constrain the primary drivers of INP activity in the Southern Ocean.

The INP distribution we present here provides a valuable tool for the community. Running a fully-coupled aerosol microphysics scheme adds considerable computational expense to simulations, hence cloud-ice microphysics schemes are not commonly coupled to aerosol fields. For simplicity, temperature-dependent parameterizations are often used to calculate INP concentrations (e.g., Meyers et al. (1992); Cooper (1986)), however they cannot represent spatial or temporal variability. Others are more complex and are dependent on concentrations of large (over 0.5 μm diameter) non-SSA particles (DeMott et al., 2010) or mineral dust particles (DeMott et al., 2015). Our model builds on these complex parameterizations, and provides a 4D climatology of INP concentrations for use in global and regional modelling studies, as well as other relevant applications. Though we do not fully understand the mechanisms involved, the INP distribution using the fertile soil parameterization given by O'Sullivan et al. (2014) is able to best represent the INP measurements. This informs us on the direction of future model developments, with a need to represent dust aerosol emissions from both desert and fertile soils, rather than just a single class.

*Code and data availability.* All data relevant for producing the figures and values and the necessary python scripts are available on Zenodo at https://zenodo.org/records/11186250 (Herbert, 2024).

*Author contributions.* BM and KC designed the study. ASM and KP configured the model and ASM ran the simulations. RH and ASM developed the analysis codes and performed the analysis, with assistance from KP and DG. BM, KC and SA contributed to discussions and direction of analysis. RH and ASM led the manuscript preparation with contributions from all co-authors.

*Competing interests.* At least one of the co-authors is a member of the editorial board of Atmospheric Chemistry and Physics. The authors have no other competing interests to declare.

*Acknowledgements.* This research has been supported by the Natural Environment Research Council (grant no. NE/T00648X/1). This work used Monsoon2, a collaborative High-Performance Computing facility funded by the Met Office and the Natural Environment Research Council, and JASMIN (http://jasmin.ac.uk/), the UK collaborative data analysis facility. DPG acknowledges support from the Centre for Environmental Modelling And Computation (CEMAC).

## Appendix A: Details of INP measurements



**Table A1.** Details of the INP measurements used for evaluation. The campaign name is included alongside the reference where available. The spatial distribution of measurements can be found in Fig. 6 and individual data sets compared against the INP model in Fig. A1

| Reference | Location | Time period | Method | N data |
|---|---|---|---|---|
| DeMott et al. (2010) AMAZE-08 | Manaus, Amazon rainforest | Feb – Mar 2008 | CFDC | 30 |
| DeMott et al. (2010) CLEX-10 | Eastern Canada | Winter 2006 and 2007 | CFDC | 54 |
| DeMott et al. (2010) INSPECT-I | Storm Peak, Central USA | Nov 2001 | CFDC | 2 |
| DeMott et al. (2010) INSPECT-II | Storm Peak, Central USA | Apr – May 2004 | CFDC | 9 |
| DeMott et al. (2010) WISP-94 | Colorado, USA | Feb – Mar 1994 | CFDC | 17 |
| Ardon-Dryer and Levin (2014) | East Mediterranean | Jan 2009 – Dec 2010 | FRIDGE-TAU cold stage | 40 |
| DeMott et al. (2016) NETCARE | Baffin Bay region | Jul 2014 | DFT flow cell | 8 |
| DeMott et al. (2016) MAGIC | Eastern Pacific | Jul 2013 | Droplet freezing assay | 12 |
| DeMott et al. (2016) ICE-T VI | Virgin Islands | Jul 2011 | Droplet freezing assay | 12 |
| DeMott et al. (2016) ICE-T PR | Puerto Rico | Jul 2011 | Droplet freezing assay | 8 |
| Mason et al. (2016) Alert | Arctic | Mar – Jul 2014 | DFT flow cell | 3 |
| Mason et al. (2016) USA | Kansas | Oct 2014 | DFT flow cell | 3 |
| Mason et al. (2016) France | Saclay | Jul – Aug 2014 | DFT flow cell | 3 |
| Mason et al. (2016) Coastal Ca | BC, Canada | Aug 2013 | DFT flow cell | 3 |
| Mason et al. (2016) Suburban Ca | BC, Canada | May 2014 | DFT flow cell | 3 |
| Mason et al. (2016) Alpine Ca | BC, Canada | Mar – Apr 2014 | DFT flow cell | 3 |
| McCluskey et al. (2018b) | Mace Head, Ireland | Aug 2015 | Various | 35 |
| Price et al. (2018) ICE-D | Cape Verde | Aug 2015 | Droplet freezing assay | 129 |
| Si et al. (2018) | Canada | Aug 2013 – Jul 2014 | DFT flow cell | 2 |
| Irish et al. (2019) | Canadian Arctic | Summer 2014 | DFT flow cell | 51 |
| Si et al. (2019) | Arctic (Alert) | Spring 2016 | DFT flow cell | 64 |
| Wex et al. (2019) Alert | Arctic | 2012 – 2016 | Cooling bath | 209 |
| Wex et al. (2019) Svalbard | Arctic | 2012 – 2016 | Cooling bath | 98 |
| Wex et al. (2019) Utqiagvik | Arctic | 2012 – 2016 | Cooling bath | 177 |
| Wex et al. (2019) Villum | Arctic | 2012 – 2016 | Cooling bath | 62 |
| Jiang et al. (2020) | Shandong province, China | Jun 2018 | FRIDGE chamber | 3 |
| Tobo et al. (2020) | Tokyo, Japan | Aug 2016 – Jul 2017 | CRAFT cold stage | 450 |
| Sanchez-Marroquin et al. (2020) VANAHEIM-2 | Iceland | Oct 2017 | Droplet freezing assay | 65 |
| Sanchez-Marroquin et al. (2021) EMeRGe | Southeast UK | Jul – Sep 2017 | Droplet freezing assay | 57 |
| Tatzelt et al. (2021) ACE | Southern Ocean | Dec 2016 – Mar 2017 | Cooling bath | 634 |
| Harrison et al. (2022) B-ICE | Barbados | Jul – Aug 2017 | Droplet freezing assay | 32 |





**Figure A1.** Evaluation of the INP model against individual measurement data sets, detailed in Table A1. Dust INPs are represented using the K-feldspar based parameterization of Harrison et al. (2019). Marine-sourced INPs are represented with the parameterization of McCluskey et al. (2018a) applied to the surface area of SSA. The simulated concentration shown is the monthly median for the corresponding month of the measurement. The solid line corresponds to a bias of zero and the dashed lines to one and a half orders of magnitude bias. Geographical locations, shown in Fig. 6, are depicted by different marker shape/color combinations.



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
