# Peer review of "Gaps in our understanding of ice-nucleating particle sources exposed by global simulation of the UK Earth System Model"

_EGUsphere, 2024_

## Referee Comment (RC1)

General comments:

In this paper, the authors present a comprehensive study on simulating the global distribution of dust and marine ice-nucleating particles (INPs) using the UK Earth System Model (UKESM1). They incorporate two important INP types, namely dust and marine organic aerosols, into the model and evaluate the simulations against an expanded global dataset of INP measurements. Furthermore, they found that soil dust might be an important INP source which was not considered in the current model. The paper is well-structured and clearly written. The methodology is sound, and the results are presented in a logical and convincing manner. The study makes a significant contribution to understanding the global distribution of INPs and identifying potential gaps in current INP representations in climate models. The paper is a valuable contribution to the field and provides useful insights into the gaps in our understanding of INP sources and their representation in climate models. I recommend publication after addressing the following comments:

**Model evaluation using satellite observations:**
The authors demonstrate the model's skill in capturing the spatial and temporal variability of INP concentrations through a thorough comparison with observations from diverse geographical locations and seasons. However, as the authors acknowledge, the current INP observations are primarily short-term campaign measurements, lacking long-term continuous time series. Relying solely on these ground-based INP observations may not be sufficient for a comprehensive evaluation of the model's performance. To address this limitation, I suggest that the authors consider utilizing satellite remote sensing data to further validate their model results. Satellites can provide valuable information on cloud phase and cloud-top temperature, which can be used to infer the threshold temperature for immersion freezing INPs (Carlsen et al., 2022). This satellite-based diagnostic approach can help identify potential INP sources on a global scale, complementing the ground-based observations. The authors should discuss the advantages and limitations of using satellite data for model validation in their discussion section.

**Potential role of anthropogenic pollution in the Northern Hemisphere mid to high latitudes:**
The most intriguing finding of this study is the discrepancy between the observed and modeled ice-nucleating particle (INP) concentrations in the mid to high latitudes of the Northern Hemisphere, particularly at higher temperatures. The authors suggest that soil dust may be an overlooked source of INPs in these regions. This insight opens up a new avenue for investigation and highlights the need to better understand the role of different INP sources in the global context. However, it is important to note that the mid to high latitudes of the Northern Hemisphere are also heavily influenced by anthropogenic pollution. While the contribution of anthropogenic pollutants to INP concentrations is still a matter of ongoing research and debate, there is evidence from satellite observations and field measurements that cannot be ignored. Satellite-based studies have shown that polluted regions tend to exhibit higher ice nucleation threshold temperatures, indicating a potential impact of anthropogenic emissions on ice formation processes (Zhao et al., 2019). Additionally, some recent observation found that organic aerosols in anthropogenically influenced areas can importantly contribute to INP

concentrations (Tian et al., 2022). These studies should be referenced for more comprehensive explanations of your results.

Given these findings, I suggest that the authors consider the potential role of anthropogenic pollution in their analysis and discussion. While soil dust may indeed be an important and overlooked source of INPs, the influence of anthropogenic pollutants should not be dismissed, especially in regions where their concentrations are high. The authors could strengthen their argument by addressing this aspect and discussing how the relative contributions of soil dust and anthropogenic pollutants to INP concentrations might vary spatially and temporally. By considering the potential role of anthropogenic pollution alongside soil dust, the authors can provide a more comprehensive analysis of the factors contributing to the observed discrepancies in INP concentrations in the Northern Hemisphere mid to high latitudes.

Reference:

Carlsen, T., & David, R. O. (2022). Spaceborne evidence that ice-nucleating particles influence high-latitude cloud phase. Geophysical Research Letters, 49, e2022GL098041. https://doi.org/10.1029/2022GL098041.

Zhao, B., Wang, Y., Gu, Y. et al. Ice nucleation by aerosols from anthropogenic pollution. Nat. Geosci. 12, 602–607 (2019). https://doi.org/10.1038/s41561-019-0389-4

Tian, P., Liu, D., Bi, K., Huang, M., Wu, Y., Hu, K., et al. (2022). Evidence for anthropogenic organic aerosols contributing to ice nucleation. Geophysical Research Letters, 49, e2022GL099990. https://doi.org/10.1029/2022GL099990

---

## Referee Comment (RC2)

The paper 'Gaps in our understanding of ice-nucleating particle sources exposed by global simulation of the UK climate model', by Herbert et al attempts to identify the gaps in the understanding of the ice-nucleating particles (INPs) by simulating the global distribution of dust and marine-sourced INPs over an annual cycle. The authors have put an appreciable effort into this objective. Different parameterization schemes by previous investigators are employed for this purpose and modifications are made to reduce the bias and errors. Since the science behind the ice nucleation in the atmosphere is not fully understood and ice clouds are highly relevant for their radiative and hydrologic impacts, the study topic is very appropriate within the scope of the Atmospheric Chemistry and Physics journal. However, a few concerns remain.

**General Comments**

- Does the title "UK Climate model" rightly indicate the UKESM discussed in the manuscript?
- I could not find a dedicated "Results and Discussion" in the manuscript. Rather, discussions are included in the "Conclusions". Why so? Conclusions can be separated from discussions with precise conclusions alone.
- The current study presents a global INP model relevant to immersion-mode freezing of liquid water droplets (L70). If it only considers only one nucleation process out of many complicated proposals for atmospheric ice nucleation then how the study can claim that it can unveil the gap areas in the understanding of ice-nucleating particle sources? How much the immersion-mode freezing contribute to the total observed global ice nuclei concentration? How much it varies with region? This calls for highlighting the issue in the manuscript or modifying the manuscript tile to avoid confusion to the readers. Also, how much bias/underestimation in the presented results is due to this approach?
- Another concern is related to the anthropogenic contribution or effects in the INP concentration which are not considered anywhere nor discussed in section 5. Many regional studies have highlighted the role of anthropogenic dust and even the role of soot and BC as INPs. However, those discussions are missing in the current study except for a few references in section 4.2.2. The authors shall mention the studies from the Indian sub-continent which indicate the influence of pollution/anthropogenic in ice nuclei concentrations. Only a few or NO measurements from the most populated regions of the globe especially in the tropics are used in the current study (Fig 6). In this regard, how well the current study can justify the claim as a global evaluation? Table A1 does not fit for a global evaluation.
- Connected to the above concern, the statement made in the conclusions (L671) is misleading. INP concentration is maximum along the tropics especially where the human population is also high (Fig 4c) and the associated influence in hydrological cycle is also relevant. Considering this, how the statement in L671 can be generalized? This is a significant scientific aspect especially in the context of the dehydration of the tropical tropopause layer due to ice nucleation (Ref: works of Eric J. Jensen from NASA). Thus, the INP studies are as relevant in the tropics as much in mid-latitudes or polar circles.

**Other Comments**

- Please specify the references for equations 1 and 2
- Please re-write the sentence in L187 (..dust concentrations so poor dust optical depth..). Could not understand
- Please give the references for the statement in L434-436.
- Please re-write the sentence in L653-655. Could not understand

---

## Author Comment (AC1)

**Response to referee comments (RC1 and RC2) by Herbert and coauthors.**

Dear editor, we would like to thank the two referees for providing us with comments on our manuscript. We have produced a revised manuscript that we believe satisfies all concerns and suggestions raised by the referees. The main modification we have made is to add black carbon ice-nucleating particles (INPs) to our INP model in order to account for anthropogenic activities (alongside marine and dust sources). Though some of the statistics slightly change, our conclusions remain unaffected (and is consistent with our previous modelling of black carbon INP). We feel that this addition has greatly improved the INP model and we thank both referees for this suggestion. Below we respond to each comment in turn, with our response indented for clarity. Where appropriate we have included the revised sections of text.
* * *
**Response to Referee #1 (RC1)**

In this paper, the authors present a comprehensive study on simulating the global distribution of dust and marine ice-nucleating particles (INPs) using the UK Earth System Model (UKESM1). They incorporate two important INP types, namely dust and marine organic aerosols, into the model and evaluate the simulations against an expanded global dataset of INP measurements. Furthermore, they found that soil dust might be an important INP source which was not considered in the current model. The paper is well-structured and clearly written. The methodology is sound, and the results are presented in a logical and convincing manner. The study makes a significant contribution to understanding the global distribution of INPs and identifying potential gaps in current INP representations in climate models. The paper is a valuable contribution to the field and provides useful insights into the gaps in our understanding of INP sources and their representation in climate models. I recommend publication after addressing the following comments:

**Model evaluation using remote-sensing data:**

The authors demonstrate the model's skill in capturing the spatial and temporal variability of INP concentrations through a thorough comparison with observations from diverse geographical locations and seasons. However, as the authors acknowledge, the current INP observations are primarily short-term campaign measurements, lacking long-term continuous time series. Relying solely on these ground-based INP observations may not be sufficient for a comprehensive evaluation of the model's performance. To address this limitation, I suggest that the authors consider utilizing satellite remote sensing data to further validate their model results. Satellites can provide valuable information on cloud phase and cloud-top temperature, which can be used to infer the threshold temperature for immersion freezing INPs (Carlsen et al., 2022). This satellite-based diagnostic approach can help identify potential INP sources on a global scale, complementing the ground-based observations. The authors should discuss the advantages and limitations of using satellite data for model validation in their discussion section.

> We thank the referee for the suggestion. We agree that satellite observations are useful for evaluating models on global scales. However, for our INP evaluation we require accurate measurements of INPs, which cannot currently be estimated by remote sensing platforms. Several recent studies, including Carlsen and David (2022), have used satellite observations of cloud properties to infer changes in INP concentrations. However, there are many highly uncertain microphysical processes that link the primary production of ice via INPs to observable bulk cloud properties. For example, highly uncertain secondary ice production processes enhance the ice particle concentration in some regimes, whereas in longer lived clouds INPs are likely

depleted through initiation of precipitation. Thus, the association between INPs and sea-ice cover presented in Carlsen and David (2022) could also reflect differences in cloud dynamics and microphysics rather than INPs. In our study we are focusing on a bottom-up approach whereby we first constrain the INP concentrations. To achieve this, we have collated 31 INP measurement datasets that span all seasons and all continents. These include regions close to important INP sources (Cape Verde and The Middle East), remote regions (the Arctic and Southern Ocean), and regions impacted by anthropogenic aerosol (North America, Europe, East Asia). This provides us with an excellent dataset for making a robust evaluation.

For these reasons we do not believe the addition of satellite observations would be of great benefit to our evaluation and could even be quite misleading. However, we have added some relevant text in the conclusions section that discusses the use of remote observations for data validation.

*"For this to be possible we require long-term INP measurements that span these timescales. Remote-sensing platforms provide an excellent opportunity to make such measurements, yet complex cloud dynamical and microphysical processes may obscure any link between INP availability and the resulting bulk-scale cloud properties (Korolev et al., 2017; Morrison et al., 2020). This uncertainty associated with cloud microphysical processes restricts the use of remote-sensing observations for quantification of INP concentrations (e.g., Carlsen and David (2022); Zhang et al. (2012)). An alternative direction is extending current INP measurements to provide greater spatial coverage and longer sampling periods."*

**Potential role of anthropogenic pollution in the Northern Hemisphere mid to high latitudes:**

The most intriguing finding of this study is the discrepancy between the observed and modeled ice-nucleating particle (INP) concentrations in the mid to high latitudes of the Northern Hemisphere, particularly at higher temperatures. The authors suggest that soil dust may be an overlooked source of INPs in these regions. This insight opens up a new avenue for investigation and highlights the need to better understand the role of different INP sources in the global context. However, it is important to note that the mid to high latitudes of the Northern Hemisphere are also heavily influenced by anthropogenic pollution. While the contribution of anthropogenic pollutants to INP concentrations is still a matter of ongoing research and debate, there is evidence from satellite observations and field measurements that cannot be ignored. Satellite-based studies have shown that polluted regions tend to exhibit higher ice nucleation threshold temperatures, indicating a potential impact of anthropogenic emissions on ice formation processes (Zhao et al., 2019). Additionally, some recent observation found that organic aerosols in anthropogenically influenced areas can importantly contribute to INP concentrations (Tian et al., 2022). These studies should be referenced for more comprehensive explanations of your results.

Given these findings, I suggest that the authors consider the potential role of anthropogenic pollution in their analysis and discussion. While soil dust may indeed be an important and overlooked source of INPs, the influence of anthropogenic pollutants should not be dismissed, especially in regions where their concentrations are high. The authors could strengthen their argument by addressing this aspect and discussing how the relative contributions of soil dust and anthropogenic pollutants to INP concentrations might vary spatially and temporally. By considering the potential role of anthropogenic pollution alongside soil dust, the authors can provide a more comprehensive analysis of the factors contributing to the observed discrepancies in INP concentrations in the Northern Hemisphere mid to high latitudes.

We thank the referee for this suggestion, and we agree that it would be beneficial to consider the role of anthropogenic INPs in our model. We have now added a third INP source which we use as a proxy for anthropogenic INPs. For this we have used the simulated black carbon (BC) aerosol from GLOMAP and applied an ns parameterization from literature. BC in GLOMAP is emitted from wildfires, biofuels, and fossil fuels. The $n_s(T)$ parameterization is taken from Schill et al. (2020) and is representative of carbonaceous material including elemental soot, wildfire emissions, and diesel exhaust. The contributions of BC INPs from carbonaceous sources were added to our INP model and all analysis was repeated. We have expanded Figure 4 to show the global distribution of BC INPs at -20C and 500m, and their contribution to the total INP concentration. We find BC INPs present over the biomass burning regions and polluted regions at concentrations comparable to marine INP but primarily over land where dust INP concentrations tend to be larger. Due to this we do not see a strong signal from BC INPs over the polluted regions of Europe, South Asia, and East Asia, and instead see dust INPs dominating the total concentration. Therefore, this does not change our conclusions or results, but we include the BC contribution in all of the analysis. We do find that including BC INPs increases the simulated representation of INPs over the subtropics where, as the referee points out, is where anthropogenic pollution is most prevalent. We are therefore grateful for the referee suggesting this addition, and we feel it strengthens the INP model.

There are several key additions to the manuscript. These include an introduction to the new INP type and treatment (Sect 2.4), an extension of Figure 4 to include BC INPs, replotting all necessary figures where simulated INP concentrations were shown, updating statistics (RMSE, MBE) in all relevant figures and tables, a discussion of the BC INP concentrations and contributions in Sect 4.1, and additional discussion on the role of BC/anthropogenic INPs (and associated caveats) in the Conclusions. The optimized dust representation (Sect 5.4) was not affected by the addition of BC. We have included the relevant references as suggested by the referee. Our concluding paragraph on black carbon / anthropogenic INPs is shown below:

*"Black-carbon INPs are unlikely to be globally important but may influence regional concentrations. Away from these regions the contribution is negligible. Over East Asia BC INPs from bio-fuel and fossil-fuel emissions represent a maximum of ∼ 10% of total INPs measurable at the surface, whereas in South Asia, Europe, and North America the contribution is < 5%. This is consistent with other global modelling studies (Vergara-Temprado et al., 2018a; Schill et al., 2020; Kanji et al., 2020) but suggests there may be a role for BC INPs on local scales and in synoptic conditions with low dust concentrations or in remote marine environments (Thomson et al., 2018). This is consistent with some observational studies that suggest non-volatile organics contribute to the INP population (Tian et al., 2022; Zhao et al., 2019) but is inconsistent with others (Adams et al., 2020; Zhang et al., 2022; Yadav et al., 2019; Bi et al., 2019). The community would benefit from more compositional information on anthropogenic aerosols and the ice-nucleating activity of each component. In this study we only consider black carbon aerosols yet other components, such as anthropogenic-sourced organic aerosol (Tian et al., 2022) or traffic-influenced dust emissions (Chen et al., 2024), may also act as locally important sources of INPs in polluted regions."*

**Response to Referee #2 (RC2)**

The paper 'Gaps in our understanding of ice-nucleating particle sources exposed by global simulation of the UK climate model', by Herbert et al attempts to identify the gaps in the understanding of the ice-nucleating particles (INPs) by simulating the global distribution of dust and marine-sourced INPs over an annual cycle. The authors have put an appreciable effort into this objective. Different parameterization schemes by previous investigators are employed for this purpose and modifications are made to reduce the bias and errors. Since the science behind the ice nucleation in the atmosphere is not fully understood and ice clouds are highly relevant for their radiative and hydrologic impacts, the study topic is very appropriate within the scope of the Atmospheric Chemistry and Physics journal.

However, a few concerns remain.

General Comments

Does the title "UK Climate model" rightly indicate the UKESM discussed in the manuscript?

> We have changed the model reference in the title from "*UK climate model*" to "*UK Earth System Model*".

I could not find a dedicated "Results and Discussion" in the manuscript. Rather, discussions are included in the "Conclusions". Why so? Conclusions can be separated from discussions with precise conclusions alone.

> In our manuscript we chose to include the bulk of the discussion amongst our results, with the conclusions section largely dedicated to summarising the study, highlighting advances and caveats, and outlining future direction. We only include minor discussion in the conclusions section. This is consistent with ACP guidelines (https://www.atmospheric-chemistry-and-physics.net/policies/guidelines_for_authors.html). In light of the referee's comment we have included a paragraph at the end of the introduction (Sect. 1) to present the layout of the manuscript, within which we highlight that ongoing discussion is included within the results sections (Sects. 4 - 5). This reads as follows:
>
> *"The manuscript is prepared as follows. Section 2 introduces the climate model and the methods by which we represent the three classes of INPs (dust, marine-sourced, and BC). In Sect. 3 we establish the efficacy of our model by evaluating the simulated seasonal cycles of dust, SSA, and PMOA concentrations against ambient aerosol measurements. Section 4 first introduces the global distribution of INPs simulated by our model then evaluates the simulated INP concentrations against a dataset of INP measurements; the INP model evaluation includes global and regional evaluations (Sects. 4.2.1 – 4.2.2), and a focus on marine-sourced INP in the Southern Ocean (Sect. 4.2.3). Following the outcomes of the evaluation we next focus on identifying missing INPs in our model; Sect. 5 introduces a number of possible sources and determines which are most likely to be contributing to our model bias. In Sect. 6 we summarize the main findings of our study and bring the results together to inform the direction of future studies."*

The current study presents a global INP model relevant to immersion-mode freezing of liquid water droplets (L70). If it only considers only one nucleation process out of many complicated proposals for atmospheric ice nucleation then how the study can claim that it can unveil the gap areas in the understanding of ice-nucleating particle sources? How much

the immersion-mode freezing contribute to the total observed global ice nuclei concentration? How much it varies with region? This calls for highlighting the issue in the manuscript or modifying the manuscript tile to avoid confusion to the readers. Also, how much bias/underestimation in the presented results is due to this approach?

> We thank the referee for this comment. In this study we focus on INPs that are relevant for immersion-mode heterogeneous ice nucleation within supercooled cloud droplets, which occur within warm and mixed-phase clouds. Other freezing modes relevant for these cloud types are contact freezing and condensation freezing, which are thought to be less important compared to the immersion mode (Murray et al., 2012), and homogeneous freezing, which occurs in the absence of INPs (or without the INP playing a role) at below ~-35 C (Herbert et al., 2015). For colder clouds at higher altitudes (e.g., cirrus) nucleation occurs below water saturation, but evidence suggests that the particle types relevant for immersion-mode freezing are not necessarily relevant for deposition-mode freezing (Holden et al., 2021). Therefore, we focus our study on the aerosol particles that are responsible for primary ice production via immersion mode heterogeneous ice nucleation in mixed-phase clouds.

> The measurements we compare to were all made using instruments that measure the INP concentration at or above water saturation, and therefore relevant for the immersion mode. This allows us to use the INP measurements as a benchmark for our INP model. Hence, we do not introduce any bias into our model using this approach.

> To avoid possible confusion, we have amended the manuscript as follows.

> We have clearly stated the focus of this study in the abstract: *"Here we use the UK Earth System Model to simulate the global distribution of dust, marine-sourced, and black carbon INPs suitable for immersion-mode freezing of liquid cloud droplets over an annual cycle."*

> In the opening paragraph of the introduction we state why we focus on this mode of heterogeneous ice nucleation and provide appropriate references: *"This process is referred to as immersion-mode heterogeneous ice nucleation and is thought to be the dominant heterogeneous freezing mechanism in mixed-phase clouds (Murray et al., 2012; Burrows et al., 2022). Immersion-mode freezing generally operates between 0 ∘C and about -37 ∘C (Murray et al., 2012), below which homogeneous ice nucleation dominates (Koop and Murray, 2016; Herbert et al., 2015)"*

> We have clarified that the literature we refer to is for immersion-mode INPs relevant or mixed-phase clouds in the introduction: *"Laboratory, field, and modelling studies have identified several components of aerosol particles that make them effective immersion-mode INPs"* and *"Global and regional models have been used to estimate the concentrations and seasonal cycles of INPs relevant for mixed-phase clouds from different sources"*.

> In the opening paragraph of the conclusions section we include: *"In this study we present the first global simulations of ice-nucleating particles (INPs) relevant for immersion-mode freezing of cloud droplets…"* and *"Multi-decadal simulations were run in order to quantify a mean annual cycle of INPs relevant for mixed-phase clouds."*

Another concern is related to the anthropogenic contribution or effects in the INP concentration which are not considered anywhere nor discussed in section 5. Many regional

studies have highlighted the role of anthropogenic dust and even the role of soot and BC as INPs. However, those discussions are missing in the current study except for a few references in section 4.2.2. The authors shall mention the studies from the Indian sub-continent which indicate the influence of pollution/anthropogenic in ice nuclei concentrations.

> This concern was also raised by Referee #1. We have now included BC INPs as a proxy for anthropogenic INPs and have amended the manuscript in response. This includes reference to Yadav et al. (2019) who report measurements from various sites across Northern India and find no clear role of pollution. Please see the response to Referee #1 for full details.

Only a few or NO measurements from the most populated regions of the globe especially in the tropics are used in the current study (Fig 6). In this regard, how well the current study can justify the claim as a global evaluation? Table A1 does not fit for a global evaluation.

> As the referee points out there is a considerable lack of in-situ INP measurements from the most populated regions of the world, however we have included measurements from China and Japan. Another important point is that we need measurements in 'background' sites that are representative of all INP inputs to the atmosphere. In contrast, sites specifically in source regions, that are often much smaller than our grid boxes, are very useful for establishing source strengths but are not necessarily what we need for a global model assessment. In our conclusions we discuss the importance of additional measurements, especially on longer timescales, that are needed to continue to improve our evaluating capabilities. The evaluation dataset we use includes 31 datasets that span a range of seasons and continents. This is by no means exhaustive but it represents a substantial database. We refer to the evaluation as "*global*" as we use the entire global dataset together with the global distribution of simulated INP. We additionally subset this into regional evaluations. We find the same result in both the global and regionally subset comparisons, which strongly suggests there is a consistent bias being introduced across the globe.

Connected to the above concern, the statement made in the conclusions (L671) is misleading. INP concentration is maximum along the tropics especially where the human population is also high (Fig 4c) and the associated influence in hydrological cycle is also relevant. Considering this, how the statement in L671 can be generalized? This is a significant scientific aspect especially in the context of the dehydration of the tropical tropopause layer due to ice nucleation (Ref: works of Eric J. Jensen from NASA). Thus, the INP studies are as relevant in the tropics as much in mid-latitudes or polar circles.

> Ice nucleation under conditions relevant to the tropical tropopause is not within scope of this study. In response to the referee's previous comment on nucleation modes we have revised the manuscript throughout to clarify the scope of the study. Although Figure 4c shows large INP values these are only shown for aerosol size distributions taken from 500 m and a reference freezing temperature of -20 C. With this we are able to see the full global distribution of INPs that would be comparable to those being measured by instruments. This is important as this is the means by which we evaluate the model. In the following figure (Figure 5) we use model fields of temperature and aerosol size distributions to gain a more realistic picture of where INPs are actually likely to be impacting ice production. In the tropics altitudes of over 9 km and temperatures lower than -30 C are required for INP concentrations exceeded 1 $L^{-1}$. In these conditions homogeneous freezing will soon dominate the production of ice in deep convective clouds, and in the absence of liquid water ice production will occur via the deposition mode. For these reasons we do not believe

immersion mode freezing is as important here as it is in mid-latitude and polar regions. However, we do agree that the statement in the conclusions could be misleading. We have therefore expanded the statement to discuss the role of INPs in the tropics, this includes reference to works concerning the dehydration of the tropopause, and the impact of INP perturbations to tropical deep convective clouds.

We have added the following text into Section 4.1 when discussing the profiles of ambient INP concentrations:

"*In the tropics (30 S – 30 N) INPs are active at mid-tropospheric altitudes (> 6 km) where aerosol concentrations tend to be lower. Here INPs may play an important role in influencing radiative properties of deep convective clouds (Hawker et al., 2021).*"

The following has been included in the paragraph summarising the distribution of simulated INPs within the conclusions section:

"*Dust uplifted to high altitudes contributes to the availability of high-altitude INPs that may influence high-cloud radiative properties and humidification of the stratosphere (Hawker et al., 2021; Jensen et al., 2013).*"

We have also rewritten and expanded the statement highlighted by the referee. The revised text reads as follows:

"*Therefore, we recommend in-situ INP measurements from within the lower atmosphere are targeted within these latitudes to gain a representative dataset for evaluation. We also simulate INPs in the tropical mid-troposphere (~10 km) that may influence tropical high-cloud radiative properties, yet there is a distinct lack of INP measurements at these altitudes. We recommend targeting these altitudes for sampling of air and INP analysis.*"

Other Comments

Please specify the references for equations 1 and 2

We have added relevant references as requested.

Please re-write the sentence in L187 (..dust concentrations so poor dust optical depth..). Could not understand

The sentence in question is "For our INP model we are primarily interested in representing the dust concentrations so poor dust optical depth performance is of secondary importance to our analysis."

We have rewritten the sentence as follows:

"*For our INP model we are primarily interested in representing the dust concentrations. Therefore, the lack of skill in reproducing observed dust optical depths is of secondary importance to our analysis.*"

Please give the references for the statement in L434-436.

The statement in question is "Some dust-producing regions such as the Sahara Desert have typically low TOC contents (<1 kg C m$^{-2}$), whereas other dust emission

regions such as parts of North America and central Russia have relatively high TOC content (>10 kg C m$^{-2}$)." These values are extracted from the figure directly above this statement and are preceded by a sentence introducing the figure.

Please re-write the sentence in L653-655. Could not understand

[revised manuscript text omitted]